# FACE: Evaluating Natural Language Generation with Fourier Analysis of Cross-Entropy

**Zuhao Yang**[1]*, **Yingfang Yuan**[2]*, **Yang Xu**[3]*†, **Shuo Zhan**[1], **Huajun Bai**[4], **Kefan Chen**[2]

[1] School of Computer Science and Engineering, Nanyang Technological University
[2] School of Mathematical and Computer Sciences, Heriot-Watt University
[3] Department of Computer Science, Southern University of Science and Technology
[4] Genify

## Abstract

Measuring the distance between machine-produced and human language is a critical open problem. Inspired by empirical findings from psycholinguistics on the periodicity of entropy in language, we propose FACE, a set of metrics based on _F_ourier _A_nalysis of the estimated _C_ross-_E_ntropy of language, for measuring the similarity between model-generated and human-written languages. Based on an open-ended generation task and the experimental data from previous studies, we find that FACE can effectively identify the human-model gap, scales with model size, reflects the outcomes of different sampling methods for decoding, correlates well with other evaluation metrics and with human judgment scores.

## 1 Introduction

The concept of _entropy_ from Information Theory is broadly applied in Natural Language Processing (NLP) technology and computational linguistic studies. The most notable example is the use of _cross-entropy_ in training and evaluating language models, where the exponentiation of cross-entropy, perplexity, is adopted to measure models' performance in next-word (or masked-word) prediction task. However, low perplexity alone does not guarantee good performance in language generation tasks, which not only depend on model sizes but are also closely related to the sampling techniques used in _decoding_ stage. The complexity of the generation task makes it especially important to have different metrics that can reflect the generation quality from multiple angles. One particular perspective is that the language generated from a good model should have a similar distribution of words/tokens as in the "natural" human language.

Recent advances in psycholinguistics put forward new directions for developing more sophisticated metrics other than Zipf's coefficient. In particular, studies on temporal and spectral patterns in dialogue [7, 47] reveal that cross-entropy (or referred to as surprisal, information density in the psycholinguistics literature) changes _periodically_ in natural language, which points out the potentials of using fine-grained transformation of cross-entropy to quantify the differences in language data (see Section 3 for a detailed review). It motivates the basic idea of this study: Can we effectively quantify the _periodical_ pattern of the cross-entropy, and use it as an indicator to distinguish human and model-generated languages?

We summarize our contributions as follows: 1. We propose a set of metrics based on the frequency spectra obtained from the Fast Fourier Transform (FFT) of the cross-entropy sequences of language data, named FACE (_F_ourier _A_nalysis of _C_ross-_E_ntropy). 2. We empirically show FACE's performance on identifying human-model gap and how it scales with model sizes in Section 4.1. 3. We explore

---

*Equal contribution, in random order
†Corresponding author, email: xuyang@sustech.edu.cn

37th Conference on Neural Information Processing Systems (NeurIPS 2023).

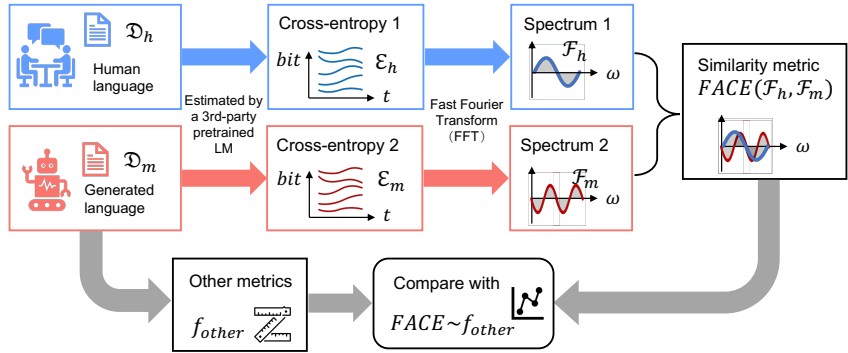

Figure 1: Overall workflow of this study.

FACE's correlations with sampling methods and human evaluation in Section 4.2 and Section 4.3, respectively. 4. We validate the statistical soundness of FACE in Section 4.4. 5. We discuss an intuitive interpretation of the metrics and how it reflects the characteristics of language use in Section 4.5. 6. Implementation and experiments code are available in this public repository: https://github.com/CLCS-SUSTech/FACE.

## 2 FACE

The basic idea of FACE is to obtain the spectra of cross-entropy from different data sources (human or models) and compute their similarities. The overall workflow is shown in Figure 1, which we describe in five steps:

1. Collect the datasets for human-written and model-generated texts, $\mathcal{D}_h$ and $\mathcal{D}_m$.
2. Use a third pre-trained language model $m_{\text{est}}$ to estimate the cross-entropy of text in $\mathcal{D}_h$ and $\mathcal{D}_m$, resulting in two sequences of cross-entropy output, $\mathcal{E}_h$ and $\mathcal{E}_m$.
3. Obtain the frequency spectra for each cross-entropy sequences, $\mathcal{E}_h \Rightarrow \mathcal{F}_h$ and $\mathcal{E}_m \Rightarrow \mathcal{F}_m$.
4. Develop FACE metrics that quantify the spectral similarity between $\mathcal{F}_h$ and $\mathcal{F}_m$.
5. Evaluate FACE on different model types/sizes, sampling methods, and the correlations with other metrics for Natural Language Generation (NLG).

We describe the steps in detail from Section 2.1 to Section 2.3.

### 2.1 Estimate cross-entropy

We use a pre-trained language model $m_{\text{est}}$ as the estimator for cross-entropy, which runs in the evaluation model (no gradients produced). It takes as input a sequence of $T$ tokens, $[t_1, t_2, \ldots, t_T]$; for each position $i = 1, \ldots, T$, it predicts the probability of the next token $P(t_{i+1}|t_1, \ldots, t_i)$; the cross-entropy between this probability and the ground truth token $t_{i+1}$ is then computed, resulting in the cross-entropy sequence that consists of $T - 1$ real values $\mathcal{E} = [c_1, c_2, \ldots, c_{T-1}]$, as the first token is not predicted:

$$\mathcal{E} = [c_1, c_2, \ldots, c_{T-1}] \triangleq [-\log P(t_2|t_1), -\log P(t_3|t_1, t_2), \ldots, -\log P(t_T|t_1, t_2, \ldots, t_{T-1})] \quad (1)$$

Note that $\sum c_i = -\sum_{i=2}^{T} \log P(t_i|t_1 \ldots t_{i-1})$ is exactly the definition of negative log-likelihood loss, i.e., cross-entropy loss, for training a language model, where $c_i$ is the negative logarithm of the predicted probability for each token $t_{i+1}$. In psycholinguistic studies, this $c_i$ quantity is usually referred to several different terms, including *surprisal* [16, 17], *information density* [25, 20, 48], and *entropy* [13, 14, 46, 47], each of which has a specific theoretical flavor. There have been debates over the justifiability of using "entropy" to denote the negative log-likelihood, because it is not a weighted summation as originally defined in [37]. Albeit, we decide to use *cross-entropy* as it is the most broadly communicated term and we believe it will not cause confusion as its mathematical form is clearly defined. Apparently, the choice for $m_{est}$ will influence the next steps, because better

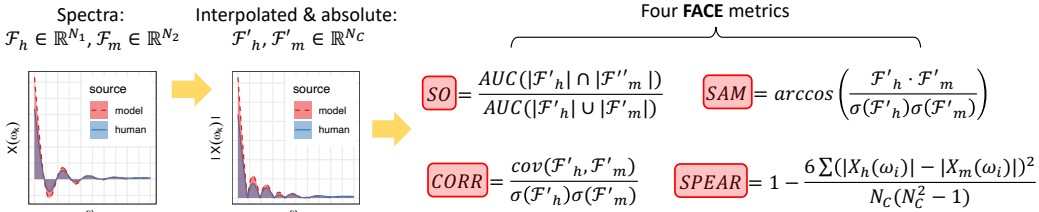

Figure 2: Definitions of four FACE metrics.

language models produce lower perplexity scores, that is, lower cross entropy. Therefore, we discuss how different choices for $m_{est}$ affect our metrics in Section 4.4.

## 2.2 Fast Fourier transform

We treat the estimated cross-entropy sequence $[c_1, \ldots, c_{T-1}]$ as a finite discrete signal in the time domain, where the sampling interval is approximated with the average duration of one token. With this simplified assumption, we find that the discrete Fourier transform (DFT) is the most suitable spectral analysis tool [39] [3]. The formula for DFT is as follows:

$$X(\omega_k) \triangleq \sum_{n=0}^{N-1} x(t_n) e^{-j\omega_k t_n}, \ k = 0, 1, \ldots, N-1 \qquad (2)$$

in which $x(t_n)$ is the signal at time $t_n$, corresponding to the $n$-th cross-entropy value $c_n$ ($n = 1 \ldots, T-1$ and $N \triangleq T-1$). $X(\omega_k)$ is a complex number that reflects the magnitude (strength) of the $k$-th frequency component $\omega_k = 2\pi k/N$. In practice, DFT is implemented with an efficient algorithm known as Fast Fourier Transform [5] that runs in $O(n \log n)$ time.

We compared two methods, periodogram and vanilla FFT. The periodogram approach computes the Fourier transform after applying auto-correlation and time-averaging windows to the signal for de-noising purposes [43]. However, we think de-noising is inappropriate because our "signal" is a time series of cross-entropy, whose value reflects the sampling result at each time step from a large. Auto-correlation or time averaging will remove the distinctiveness of rare tokens. Therefore, we use vanilla FFT and take the *real* part of $X(\omega_k)$ to represent the magnitude spectrum for the frequency component $\omega_k$, which is written as $X(\omega_k)$ for brevity.

For an input cross-entropy sequence $\mathcal{E} = [c_1, \ldots, c_{T-1}]$ obtained from Section 2.1, the resulting frequency spectrum can be represented as a list of tuples of the same length, $\mathcal{F} = [\langle \omega_1, X(\omega_1) \rangle, \ldots, \langle \omega_{T-1}, X(\omega_{T-1}) \rangle]$, where $[\omega_1, \ldots, \omega_{T-1}]$ are the $T-1$ sample frequencies, and $[X(\omega_1), \ldots, X(\omega_{T-1})]$ are the corresponding magnitudes.

## 2.3 Spectral similarity metrics

We develop four metrics to measure the similarity between spectra $\mathcal{F}_h$ and $\mathcal{F}_m$: Spectral Overlap (*SO*), Spectrum Angle Mapper (*SAM*) [6], Pearson's correlation (*CORR*), and Spearman's correlation (*SPEAR*), as summarized in Figure 2. Before computing the metrics, two spectra $\mathcal{F}_h$ and $\mathcal{F}_m$ which are of different lengths $N_1$ and $N_2$, are first interpolated to the same length: $\mathcal{F}_h \in \mathbb{R}^{N_1} \Rightarrow \mathcal{F}'_h \in \mathbb{R}^{N_C}$, $\mathcal{F}_m \in \mathbb{R}^{N_2} \Rightarrow \mathcal{F}'_m \in \mathbb{R}^{N_C}$. Here, $N_C$ is the maximum length of the spectrum in our data. Thereafter, the computation of the subsequent metrics can commence.

**Spectral Overlap (*SO*)** is inspired by the power spectrum overlap proposed in [28], which is used in [47] for measuring the spectral similarity between dialogue participants. The frequency magnitudes in $\mathcal{F}'_h$ and $\mathcal{F}'_m$ are converted to absolute values, i.e., $X(\omega_k) \Rightarrow |X(\omega_k)|$, and then compute the Area-Under-Curve (AUC) for the interaction $\mathcal{F}'_h \cap \mathcal{F}'_m$ and the union $\mathcal{F}'_h \cup \mathcal{F}'_m$, respectively. *SO* is defined as the ratio of the two: $SO = \text{AUC}(\mathcal{F}'_h \cap \mathcal{F}'_m)/\text{AUC}(\mathcal{F}'_h \cup \mathcal{F}'_m)$. The procedure of converting

---

[3]`https://ccrma.stanford.edu/~jos/sasp/Fourier_Transforms_Continuous_Discrete_Time_Frequency.html`

to absolute values is indispensable, since negative values in $X(\omega_k)$ will result in negative AUCs. *SO* has the range $[0, 1]$, and a higher value indicates a stronger resemblance between the two spectra.

**Spectrum Angle Mapper (*SAM*)** calculates the angles between $\mathcal{F}'_h$ and $\mathcal{F}'_m$, treating them as two vectors in a space [22]. The angle is measured in radians, which is calculated by the inverse function $\arccos(\mathcal{F}'_h \cdot \mathcal{F}'_m / ||\mathcal{F}'_h|| \cdot ||\mathcal{F}'_m||)$, producing a value within $[0, \pi]$. We understand *SAM* is equivalent to the cosine similarity score, which is more commonly-used in NLP, but here we just follow the conventions in [22, 2]. A smaller *SAM* value indicates a greater similarity between $\mathcal{F}'_h$ and $\mathcal{F}'_m$.

**Pearson's correlation (*CORR*)** can also be leveraged to measure spectral similarities as discussed in [22]. $CORR = cov(\mathcal{F}'_h, \mathcal{F}'_m)/\sigma(\mathcal{F}'_h)\sigma(\mathcal{F}'_m)$, with a $[-1, 1]$ range. A positive *CORR* value indicates high similarity (negative for dissimilarity), and 0 indicates weak correlation between $\mathcal{F}'_h$ and $\mathcal{F}'_m$.

**Spearman's correlation (*SPEAR*)** [41] is commonly used to assess the monotonic relationship between the comparison and reference groups and to capture the presence of non-linear associations between the two. It has not been used for spectral similarity to the best of our knowledge, but we test it in our experiments. *SPEAR* also has the range $[-1, 1]$ with meanings similar to *CORR*.

# 3 Related Work

**Entropy as a metric in psycholinguistics.** The entropy of human language has long been a research interest in computational linguistics and psycholinguistics. The entropy of written text is estimated with the average per-word negative log-probability in sentences, and then used to validate the principle of *entropy rate constancy* (ERC) in human language [13, 14]. Similar studies were conducted in dialogue [46, 32]. Entropy is also defined in probabilistic grammars to describe the capacity of a language [40], and is used to develop complexity metrics to measure the cognitive load of processing syntactic expressions [16, 24, 17]. In the line of work on language production, a different term *information density* with the same mathematical formulation is used instead of entropy. It is found that speakers reduce syntactic complexity when the information density (or entropy) is high [25, 20]. In parallel with the concept of ERC, this line of work summarizes the tendency of distributing information evenly in human language with the term *uniform information density* (UID), which is commonly used as a equivalent term as ERC, for example, in [48, 15]. In conclusion, entropy is commonly used as a metric for essential properties of human language.

**Periodical change of cross-entropy in language.** We draw inspiration from the following studies about the distribution of information in dialogue. Humans are sensitive to the *peaks* and *troughs* of entropy in speech, with evidence from human-system dialogues and crowd-sourced ratings from human judges [7]. The entropy of utterances from two speakers converge towards each other within the scope of topical segments in spontaneous dialogues [48]. They measure the entropy of utterances from two participants of a task-oriented dialogue, and have found that the frequency domain features – power spectrum overlap and phase delay – are useful predictors of task outcomes. Both works reviewed above suggest that the periodical up-and-downs of entropy are commonly observable in the human language. It naturally leads to the question of whether and to what extent model-generated language aligns with this empirical finding.

**Automatic measures for text generation.** Entropy and its related variants have already used as a metric for evaluating generated text, for instance, entropy provides good visualization for the difference between GPT2-generated text and human written ones [12]. Other than entropy, there is a rich body of existing metrics targeted on discriminating human-written text and model-generated text, which we summarize in three branches: (1) statistics-based; (2) language modeling; (3) reference-based. Table 1 gives a brief summary of these three categories, as well as our proposed frequency-based FACE.

*Statistics-based measures* compare the model-generated distribution $M$ with respect to the human-written distribution $H$ in terms of some statistic. The Zipf coefficient [30] is used in [19] to describe the distribution of word frequencies in text. Self-BLEU [52] is derived by calculating the BLEU [29] score for each generated text utilizing all other generations as references. Repetition measures the sequence-level degree of repetition on the basis of the percentage of duplicated n-grams in the generated continuations $x_{\text{cont}} \sim M$ [44]. Meanwhile, we aggregate the 2-gram, 3-gram, and 4-gram repetition rates to evaluate the lexical diversity in an inverse manner.

| Type | Metric | Measure | Definition/Approximation |
|---|---|---|---|
| Statistics | Zipf Coefficient | Unigram rank-frequency statistics | - |
| | Self-BLEU | $N$-gram diversity | - |
| | Repetition | Sequence-level percentage of repetition | $1 - \frac{\lfloor \text{unique } n\text{-grams } (\boldsymbol{x}_{\text{cont}}) \rfloor}{\text{total } n\text{-grams } (\boldsymbol{x}_{\text{cont}})}$ |
| | Diversity | Inverse of $n$-gram repetition rates ($n = 2, 3, 4$) | $\prod_{n=2}^{4}(1.0 - \text{Repetition})$ |
| Language Modeling | Perplexity | Evaluation-set perplexity | $\mathbb{E}_H[\log M(\boldsymbol{x})]$ |
| | Coherence | LM quality (cosine similarity between sentence embeddings) | $\frac{\text{EMB}(\boldsymbol{x}_{\text{pre}}) \cdot \text{EMB}(\boldsymbol{x}_{\text{cont}})}{\|\text{EMB}(\boldsymbol{x}_{\text{pre}})\| \cdot \|\text{EMB}(\boldsymbol{x}_{\text{cont}})\|}$ |
| Divergence Curve | MAUVE | Quality & diversity via the divergence frontiers | $\mathcal{C}(H, M)$ at all $\lambda \in (0, 1)$ [31] |
| Frequency Domain | FACE (this work) | Quality & diversity via the spectral similarities (four metrics) | $FACE(\mathcal{F}_h, \mathcal{F}_m)$ |
| Human Judgment | Bradley-Terry Score | Human preference via the pairwise evaluation | $P(i \text{ beats } j) = \frac{1}{1 + e^{-(\beta_i - \beta_j)/100}}$ |

Table 1: Summary of metrics (automatic & non-automatic) we employed for evaluating open-ended text generation. FACE provides a way to approximate the human-model gap in the frequency domain.

*Language modeling metrics* measure how un(certain) human text $\boldsymbol{x} \sim H$ follows the model distribution $M$, using the probability distribution $M(\boldsymbol{x})$. In our work, the perplexity is calculated upon the set of human texts to quantify how well the distribution $M$ predicts a text continuation. Coherence is approximated by cosine similarity between the sentence embeddings of prompt $\boldsymbol{x}_{\text{pre}} \sim H$ and continuation $\boldsymbol{x}_{\text{cont}} \sim M$ as proposed in [42], where the embedding $\text{EMB}(\cdot)$ is produced by the pre-trained SimCSE sentence embedding [11]. Metrics under this category never observe model-generated text samples, and hence, they cannot justify how likely $\boldsymbol{x}_{\text{cont}}$ is under the human distribution $H$.

*Reference-based measures* assess the generated text with respect to a small set of reference text, rather than calculating over the full sequence distributions. Some recent reference-based approaches encompass: (1) [4, 36, 38, 50] aim to capture distributional semantic information in high-dimensional space; (2) [51] concerns Euclidean distance between vector representations of $n$-grams and their document frequencies; (3) [31] straightforwardly computes the similarity of one learned distribution from a text generation and the other distribution of human-written text using information divergence frontiers [9, 23, 34]. Reference-based metrics are well-suited for targeted generation tasks (e.g., machine translation). Nevertheless, they become unfavorable in the open-ended generation scenario where multiple reasonable and diverse continuations are preferred.

**Non-automatic metrics.** Recent works [12, 31, 19, 26] on evaluation metrics and decoding strategies for natural language generation rely on human judgments, assuming that human annotations are the gold standard. Considering the expense of Human Unified with Statistical Evaluation (HUSE) [18], we adopt a pairwise evaluation protocol based on human preferences, to serve as a non-automatic complement of FACE metrics. We leverage the Bradley-Terry model [3] to predict the outcome of a head-to-head comparison given $n$ players with scores $\beta_1, \cdots, \beta_n$.

## 4 Experiments

**Task formulation.** Given an input text passage as prefix, the *open-ended* generation aims to produce texts that form a fluent and coherent continuation. More formally, given a sequence of $m$ tokens denoted $[x_1 \ldots x_m]$, as the **prompt**, the goal is to generate the next $n$ **continuation** tokens to form a complete sequence $[x_1 \ldots x_{m+n}]$. The continuation probability at the decoding time by conditioning on the preceding context is defined as: $P(x_{m+1} \ldots x_{m+n} \mid x_1 \ldots x_m) = \prod_{i=m+1}^{m+n} P(x_i \mid x_1 \ldots x_{i-1})$, where $P(x_i \mid x_1 \ldots x_{i-1})$ is the next-token distribution.

### 4.1 Model sizes

We consider such a text completion task in three domains: Wiki text, News, and Stories. Intuitively, the generated texts involving different domain knowledge may have different language usages and writing style, which may reflect on metrics. We generate completions from large-scale language models (LMs). In particular, we adopt three representatives of state-of-the-art pre-trained auto-regressive LMs: Generative Pre-trained Transformer 2 (GPT2) [33], Open Pre-trained Transformer LMs (OPT) [49], and BigScience Large Open-science Open-access Multilingual LM (BLOOM) [35]. We explore two sizes for each model to illustrate that our FACE metrics generalize across multiple LM families and sizes. Details regarding our task and input data are summarized in Table 2. Different

| Domain | Model | Dataset | Prompt Length | Maximum Generation Length | Number of Generations |
|---|---|---|---|---|---|
| Wiki text | GPT2/OPT/BLOOM | WikiText-103 | 35 tokens | 1024 tokens | 5000 |
| News | GPT2/OPT/BLOOM | RealNews | 35 tokens | 1024 tokens | 5000 |
| Stories | GPT2/OPT/BLOOM | WritingPrompts | varying | 1024 tokens | 5000 |

Table 2: Dataset and task summary. In our research, we set the maximum generation length to 1024 for all models on three datasets. Note that the WritingPrompts dataset [10] contains ready-to-use prompts, so the length of prompts varies. For WikiText-103 [27] and RealNews [1] datasets, we cleaned them before extracting the texts corresponding to the first 35 tokens (tokenized by GPT2Tokenizer) to form our prompt sets.

| Domain | Metric | GPT2-sm | GPT2-xl | vs. | Voting | OPT-125m | OPT-6.7b | vs. | Voting | BLOOM-560m | BLOOM-7.1b | vs. | Voting |
|---|---|---|---|---|---|---|---|---|---|---|---|---|---|
| Wiki text | Diversity (↑) | 0.733 | 0.753 | L | L | 0.645 | 0.789 | L | L | 0.533 | 0.732 | L | E |
| | Coherence (↑) | 0.595 | 0.624 | L | | 0.614 | 0.634 | L | | 0.926 | 0.819 | S | |
| | Zipf Coefficient (↓) | 0.990 | 0.975 | L | | 0.989 | 1.016 | S | | 1.092 | 0.980 | L | |
| | Self-BLEU (↓) | 0.459 | 0.424 | L | | 0.423 | 0.379 | L | | 0.280 | 0.422 | S | |
| | MAUVE (↑) | 0.677 | 0.186 | S | S | 0.169 | 0.265 | L | L | 0.517 | 0.184 | S | L |
| | SO (↑) | 0.414 | 0.406 | S | | 0.424 | 0.436 | L | | 0.426 | 0.432 | L | |
| | CORR (↑) | 0.806 | 0.781 | S | | 0.771 | 0.769 | S | | 0.675 | 0.789 | L | |
| | SAM (↓) | 0.199 | 0.213 | S | | 0.216 | 0.217 | S | | 0.258 | 0.208 | L | |
| | SPEAR (↑) | 0.022 | 0.023 | L | | 0.026 | 0.029 | L | | 0.059 | 0.023 | S | |
| News | Diversity (↑) | 0.890 | 0.897 | L | L | 0.853 | 0.876 | L | L | 0.740 | 0.870 | L | S |
| | Coherence (↑) | 0.613 | 0.640 | L | | 0.663 | 0.663 | S | | 0.897 | 0.785 | S | |
| | Zipf Coefficient (↓) | 0.961 | 0.958 | L | | 0.965 | 0.968 | L | | 0.964 | 0.966 | S | |
| | Self-BLEU (↓) | 0.619 | 0.573 | L | | 0.611 | 0.543 | L | | 0.384 | 0.501 | S | |
| | MAUVE (↑) | 0.393 | 0.281 | S | S | 0.162 | 0.130 | S | S | 0.014 | 0.095 | L | L |
| | SO (↑) | 0.424 | 0.412 | S | | 0.438 | 0.440 | L | | 0.436 | 0.437 | L | |
| | CORR (↑) | 0.757 | 0.723 | S | | 0.746 | 0.732 | S | | 0.615 | 0.733 | S | |
| | SAM (↓) | 0.224 | 0.240 | S | | 0.229 | 0.236 | S | | 0.281 | 0.234 | L | |
| | SPEAR (↑) | 0.021 | 0.019 | S | | 0.017 | 0.021 | L | | 0.048 | 0.019 | S | |
| Stories | Diversity (↑) | 0.743 | 0.785 | L | L | 0.769 | 0.875 | L | L | 0.527 | 0.830 | L | S |
| | Coherence (↑) | 0.421 | 0.420 | S | | 0.440 | 0.388 | S | | 0.880 | 0.660 | S | |
| | Zipf Coefficient (↓) | 1.097 | 1.085 | L | | 1.021 | 1.003 | L | | 0.999 | 1.058 | S | |
| | Self-BLEU (↓) | 0.617 | 0.565 | L | | 0.587 | 0.511 | L | | 0.180 | 0.455 | S | |
| | MAUVE (↑) | 0.504 | 0.121 | S | S | 0.025 | 0.013 | S | S | 0.006 | 0.008 | L | L |
| | SO (↑) | 0.411 | 0.402 | S | | 0.406 | 0.405 | S | | 0.350 | 0.418 | L | |
| | CORR (↑) | 0.813 | 0.787 | S | | 0.737 | 0.705 | S | | 0.573 | 0.772 | L | |
| | SAM (↓) | 0.195 | 0.209 | S | | 0.231 | 0.245 | S | | 0.300 | 0.214 | L | |
| | SPEAR (↑) | 0.023 | 0.022 | S | | 0.036 | 0.041 | L | | 0.050 | 0.027 | S | |

Table 3: Domain-specific generation quality with respect to different **models** (GPT2/OPT/BLOOM) and **model sizes** (large model on the left and small model on the right) using top-$k$ ($k$=50) sampling under various existing metrics, as well as proposed FACE metrics. ↑ indicates the larger the metric value, the better, whereas ↓ indicates the opposite. The *vs.* column indicates the better-performing model in each comparison, where L/S denotes the large/small model wins and E represents a tie. We applied the majority voting to determine the winner. OPT and BLOOM are postfixed with their number of parameters.

models may generate vastly different numbers of continuations in each length interval (see Appendix). To ensure the fairness of investigating the correlation between FACE and other widely-used metrics with respect to different models (with different sizes), we compute the weighted arithmetic mean for every metric across five length intervals.

The evaluation metrics we are interested in are based on various motivations and principles. Specifically, MAUVE and FACE emphasize the parallels between human and machine-produced texts, as stated in Section 3. Therefore, we group MAUVE together with four FACE metrics. To further obtain intuitive results, we utilize the voting approach to explore the correlations between these metrics on large/small models across three task domains. The results are shown in Table 3.

In our investigations, the GPT2-xl model consistently outperforms its small counterpart among statistics-based and language modeling metrics as all relevant "*vs.*" columns indicate, apart from the Coherence in the Stories domain. In the GPT2 experimental group, it is astonishing that the small model always performs better when referring to the voting results from MAUVE and FACE rows. Across three task domains, the performances of OPT and BLOOM models in two sizes differ. Large models have better overall performance, and small models only win four out of twelve comparisons by voting. Nonetheless, it is noteworthy that four FACE metrics we proposed maintain a relatively high level of consistency with MAUVE across all models. At least two FACE metrics yield the same results (in eight out of nine sets of human-model language comparisons) with MAUVE. Concretely

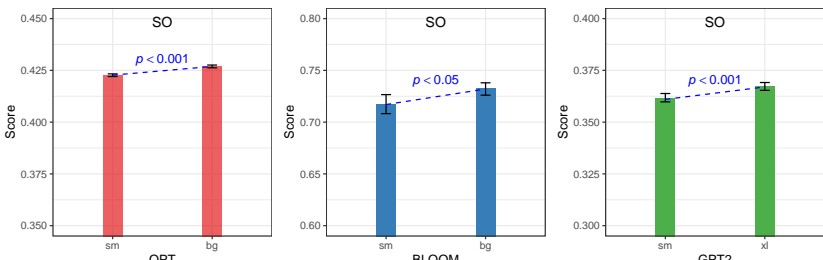

Figure 3: FACE-*SO* scores on OPT, BLOOM and GPT2 original output data. Model sizes compared: small vs. large for OPT and BLOOM; -sm vs. -xl for GPT2. Error bars represent 95% confidence intervals from bootstrap. The significant levels are based on $t$-test between the two model-size groups.

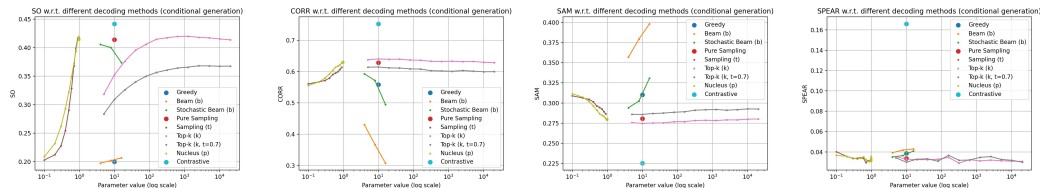

Figure 4: FACE scores (conditional generation) on original experimental data of [19] and [26]. Nine sampling methods are compared: greedy, beam search, stochastic beam search, pure sampling, temperature, top-$k$, top-$k$ with temperature, nucleus, and contrastive. Note that logarithmic normalization on parameter values as well as enlarged markers for greedy decoding, pure sampling, and contrastive decoding are adopted for better visualization effect. Best viewed when zoomed in.

speaking, *SO* and *SAM* show a higher positive correlation to MAUVE than *CORR* and *SPEAR*, given that seven out of nine voting results (marked with yellow in Table 3) are identical.

To further evaluate model sizes, we apply FACE to the original GPT2 output data (webtext) [4] generated from GPT2-sm and GPT2-xl. GPT2-xl has a higher *SO* score than GPT2-sm, which is confirmed with the $t$-test, but non-significant effects are found on the other three metrics. Combining our generation task with the original GPT2 data, we illustrate the results for *SO* in Figure 3.

To conclude, we discover three keypoints: (1) FACE is consistent with MAUVE in evaluating three different model types (two sizes for each); (2) the metrics estimating similarity between human-written and model-generated text generations (e.g., FACE, MAUVE) may produce opposite results to the text-centered metrics (e.g., Diversity, Coherence); (3) the four metrics of FACE show relatively homogeneous results, and using these metrics together helps to identify model-generated texts with a more comprehensive evaluation.

### 4.2 Sampling methods

Recent work [19, 26] has indicated three clear trends in open-ended text generation using auto-regressive LMs: (1) maximization-based decoding algorithms (e.g., beam search, greedy decoding, etc.) lead to copious repetition, while sampling with temperature may result in incoherence; (2) truncation-based sampling methods like nucleus sampling produce text with higher quality; (3) contrastive decoding outperform nucleus sampling in terms of both fluency and coherence. Accordingly, to demonstrate the effectiveness of our approach, FACE should follow the inequality: maximization-based/temperature-based $\prec$ nucleus $\prec$ contrastive in terms of the quality relationship.

Figure 4 visualizes the correlation between FACE scores and various decoding algorithms. The contrastive decoding approach yields the best performance among the four FACE metrics. It can be clearly observed that the maximization-based sampling methods behave worse than other algorithms. Moreover, adding the temperature parameter to top-$k$ sampling results in incoherent text generations,

---

[4]https://github.com/openai/gpt-2-output-dataset

| Sampling Method | Perplexity | Self-BLEU | Zipf Coefficient | Repetition | SO (↑) | CORR (↑) | SAM (↓) | SPEAR (↑) |
|---|---|---|---|---|---|---|---|---|
| Human | 12.38 | 0.31 | 0.93 | 0.28 | - | - | - | - |
| Greedy | 1.50 | 0.50 | 1.00 | 73.66 | 0.20 | 0.56 | 0.31 | 0.04 |
| Beam ($b$=16) | 1.48 | 0.44 | 0.94 | 28.94 | 0.21 | 0.31 | 0.40 | 0.04 |
| Stochastic Beam ($b$=16) | 19.20 | 0.28 | 0.91 | 0.32 | 0.37 | 0.49 | 0.33 | 0.04 |
| Pure Sampling | 22.73 | 0.28 | **0.93** | 0.22 | 0.41 | 0.63 | 0.28 | 0.03 |
| Sampling ($t$=0.9) | 10.25 | 0.35 | 0.96 | 0.66 | 0.42 | 0.61 | 0.29 | 0.03 |
| Top-$k$ ($k$=40) | 6.88 | 0.39 | 0.96 | 0.78 | 0.40 | 0.64 | 0.28 | 0.03 |
| Top-$k$ ($k$=640) | 13.82 | **0.32** | 0.96 | **0.28** | 0.42 | 0.63 | 0.28 | 0.03 |
| Top-$k$ ($k$=40, $t$=0.7) | 3.48 | 0.44 | 1.00 | 8.86 | 0.34 | 0.61 | 0.29 | 0.03 |
| Nucleus ($p$=0.95) | **13.13** | **0.32** | 0.95 | 0.36 | 0.42 | 0.63 | 0.28 | 0.03 |
| Contrastive Decoding | 14.39 | 0.54 | 1.04 | 0.24 | **0.44** | **0.75** | **0.23** | **0.17** |

Table 4: Results for comparing all sampling methods with selected parameters regarding the conditional generation. The values *closest to human scores* are **bolded**, except for our proposed FACE scores, where the *highest (for SO, CORR, and SPEAR) or the lowest (for SAM)* values are in **bold**.

which explains the gap between the red curve (top-$k$ w/o temperature) and the gray curve (top-$k$ w/ temperature). We also plot the correlation graphs of unconditional generation (in the Appendix) with fewer sampling methods involved. The trends and patterns in the visualization of unconditional generation are basically consistent with its conditional counterpart.

In Table 4, FACE scores on different decoding algorithms are summarized. FACE metrics correctly match the expected quality relationship of the sampling methods examined by assigning the best *SO* (.44), *CORR* (.75), *SAM* (.23), and *SPEAR* (.17) scores to contrastive decoding. Other evaluation metrics fail to capture the correct relationship, for example, the perplexity rates nucleus-sampled text as better than contrastive-decoded text, which is irrational suggested by Li et al. [26].

## 4.3 Human judgments

We also explore the correlation between FACE and human judgement scores, using the crowd-source dataset collected in [31] when human evaluation is available. The dataset contains model-generated continuations (by GPT2-sm, -md, -lg, and -xl with ancestral and nucleus sampling), human-written continuations using the same prefix, and the crowd-source workers' answers on which completion is more human-like, interesting, and sensible. We follow the same experimental settings and protocol to verify whether the FACE scores of the text completions correlate well with the human quality judgements by computing the Spearman's rank correlation coefficient. The results are presented in Table 5.

We observe a high and positive correlation between FACE-*SO* and human judgments scores, which outperforms five out of the six evaluation metrics reported in [31] and achieves a comparative performance against MAUVE. The remaining three FACE metrics have insignificant correlations. However, we consider human judgments to be subjective and sometimes biased. Including more fine-grained questions to perform human judgments may lead to more accurate correlation statistics. Additionally, we recomputed the correlations with human judgement scores to keep those pairs in which there are exactly one item from human and the other item from model (i.e., a subset of data used for the analysis in Table 5). As shown in *SO*-S and MAUVE-S columns, FACE-*SO* has a stronger correlation than MAUVE among two of these three dimensions.

## 4.4 Sanity tests

**Sanity test on validity.** We evaluate the validity of FACE by examining whether its scores on human-human split is lower than human-model groups – an expected result based on our assumption that the spectral difference between human's "natural" language and models' "artificial" ones should be *amplified* by FACE. Therefore, the sanity tests are conducted as follows: first, we evenly and randomly split the human data into two folds (across three domains) to serve as control groups. The FACE scores between these control folds are then computed. As for the human-to-model experimental group, we create other two folds using human data and the best model-generated data (from contrastive decoding [26]) in terms of text quality. If FACE can effectively capture the fundamental difference

| Metric | Generation Perplexity | Zipf Coefficient | Repetition | Distinct-4 | Self-BLEU | SO | MAUVE ‖ | SO-S | MAUVE-S |
|---|---|---|---|---|---|---|---|---|---|
| Human-like/BT | 0.810 | 0.833 | −0.167 | 0.738 | 0.595 | 0.881 | 0.952 ‖ | 0.357 | 0.214 |
| Interesting/BT | 0.643 | 0.524 | −0.143 | 0.524 | 0.405 | 0.762 | 0.810 ‖ | 0.524 | 0.667 |
| Sensible/BT | 0.738 | 0.690 | −0.071 | 0.595 | 0.524 | 0.786 | 0.857 ‖ | 0.995 | 0.706 |

Table 5: Spearman's rank correlation coefficients of *SO* and five other metrics with human judgments. Higher scores mean better correlation. All the numbers except the *SO*, *SO*-S, and MAUVE-S columns are sourced from [31]. "BT" denotes the Bradley-Terry score of the pairwise human evaluation, which is employed to compute the Spearman's rank correlation with the scores of other metrics. Additionally, it is important to note that the original human judgments encompass certain pairs in which both texts are generated by models, albeit different models. Therefore, we refine the original human judgment dataset to only include judgments involving both human and model-generated languages, and the results are shown in *SO*-S and MAUVE-S.

between human and model languages, then we expect to observe higher scores in control groups than in the experimental group. The results are shown in Table 6.

| | SO ($\uparrow$) | CORR ($\uparrow$) | SAM ($\downarrow$) | SPEAR ($\uparrow$) |
|---|---|---|---|---|
| h-h (wiki) | **0.45** | **0.76** | **0.22** | 0.05 |
| h-h (news) | **0.45** | **0.76** | **0.22** | 0.07 |
| h-h (stories) | **0.47** | **0.79** | **0.21** | 0.05 |
| h-m | 0.44 | 0.75 | 0.23 | **0.17** |

Table 6: Results of sanity test on FACE's validity. The top three rows are the control groups, and "h-h" stands for human-to-human folds. The last row is the experimental group, where "h-m" is for human-to-model fold. Better FACE scores are in bold. The scores in the bottom row are retrieved from Table 4.

It can be seen from Table 6 that the control groups show significantly better FACE scores than the experimental group: FACE-*SO* and FACE-*CORR* are higher in human-to-human folds, while FACE-*SAM* scores are lower. The only exception is FACE-*SPEAR*, while we will show it is a good metric in later sections. Nonetheless, these tabulated results have proved the validity of FACE in effectively capturing human-to-model spectral differences.

**Choice of estimator model.** We examine how different choices of the estimator model $m_{est}$ affect the resulting spectra, using GPT2-sm, -md, -lg and -xl as $m_{est}$, respectively. The spectra of webtext and the original GPT2 output data are computed. It is found that the spectra obtained from $m_{est}$ have different magnitudes, but their aggregated curves have the same shape (see Appendix). Therefore, the choice of $m_{est}$ will not affect FACE scores as long as the same $m_{est}$ is used for all data.

**Stationarity tests.** One of the assumptions of the Fourier transform is that the signal is *stationary* [21], i.e., the mean and variance do not change over time. We applied the Augmented Dickey-Fuller (ADF) test [8] to examine the stationarity of the cross-entropy sequences for all the human and model-generated data used in this study. The null hypothesis $H_0$ of the ADF test is non-stationarity, and thus a $p < .05$ testing result rejects $H_0$ and accepts the alternative hypothesis of stationarity in the series. We calculate the proportions of cross-entropy sequences that pass the ADF test with $p < .05$ for all model-generated and human data: 97.4% for GPT2, 92.1% for OPT, 74.5% for BLOOM, and 97.9% for human. Thus, the majority of data meets the stationarity requirement.

## 4.5 Interpretation of spectra

As the frequency spectrum reflects the key characteristics of a signal, we attempt to interpret the spectra to see if they tell how the "signals" – entropy of human and machine languages – differ. Without aggregation, the raw spectra of single cross-entropy sequence look indistinguishable between GPT2-sm, GPT2-xl, and human (see the left plot in Figure 5). By aggregating 5,000 spectra from each group and smoothing the curves, it can be seen that GPT2-xl's curve is closer to human than the GPT2-sm curve (readers can find this by zooming in the middle plot in Figure 5). Here, the smoothing is done with generalized additive models (GAMs) [45]. Results from other models are included in the Appendix.

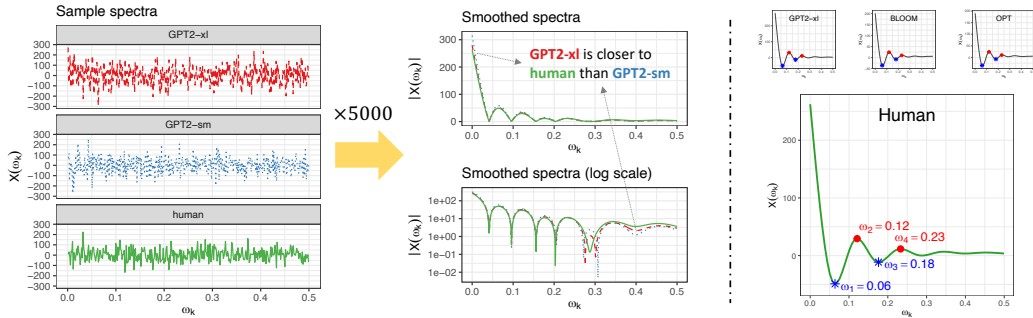

Figure 5: Intuitive observations on the spectra from GPT2 and human data (webtext). **Left**: Spectra of three randomly sampled entropy sequences from GPT2-sm, GPT2-xl, and webtext. **Middle**: Smoothed plot of 5,000 aggregated spectra with absolute values, $|X_{\omega_k}| \sim \omega_k$. **Right**: Typical smoothed plot of raw spectra $X_{\omega_k} \sim \omega_k$, with peaks and troughs annotated.

When plotted separately, the aggregated spectra from human and different models have similar shapes: First, the majority of components exist in the low-frequency range ($\omega < 0.05$). In addition, the locations of peaks and troughs are almost the same between groups. For instance, $\omega_1 = 0.06$ is the first trough, and $\omega_2 = 0.12$ is the first peak (see the right plots in Figure 5). Thus, roughly speaking, the main difference between human and model spectra is not in the locations of peak and trough frequencies but in the relative magnitudes of those frequencies.

We propose a simple way to interpret the peaks in spectra: the reciprocal of a frequency component $T_k = 1/\omega_k$ denotes the corresponding cycle in the time domain. Because the time interval (i.e., sampling interval) of an entropy sequence is not measured in *seconds* but fixed as one *token*, the measurement unit of $T_k$ is also in number of tokens. For example, the first frequency peak in Figure 5 (right plot) implies $\omega_2 = 0.12 \Rightarrow T_2 = 1/0.12 \approx 8.3$ (tokens), which approximately means that tokens of the same cross-entropy levels tend to *recur* every 8.3 tokens. This pattern is consistent in both human and model data. However, the degree of this *recurrence* can mark the difference between the human and model languages. We leave more detailed interpretations of spectra to future work.

## 5    Conclusion and Limitations

We propose FACE, a set of metrics based on the Fourier analysis of cross-entropy, which is able to distinguish human and model-generated language with satisfactory performance in the open-ended generation task. The metrics scale with model sizes; reflect the effect of various sampling methods; correlate well with other existing metrics and outperform most of them in alignment with human judgement scores. Among the four implementation methods of FACE experimented, Spectral Overlap (*SO*) has the best overall performance.

FACE is computationally efficient with easy-to-interpret output. As a method inspired by psycholinguistic studies on the predictability (entropy/surprisal/information density) of human language, we believe FACE is a good example of incorporating knowledge from different fields for better human-centered AIs. We can generally conclude that better language models can produce spectral representations of information that are more similar to human.

Our current work has several limitations: Firstly, for open-ended generation experiments (Section 4.1), a broader set of sampling methods other than top-$k$ can be used. Secondly, larger models (with more than 100 billion parameters) need to be included for more comprehensive comparisons. We will improve from these aspects in future work.

## Acknowledgement

This material is based upon work supported by the National Science Foundation under Grant No. (2105192).

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

# Appendix

## 1 Broader Impacts

FACE measures the distance between human and model-generated languages, therefore it is technically possible to be used for designing or augmenting systems that mimic humans. We acknowledge the risks of FACE (and other metrics) being utilized in applications that deliberately confuse human-authored and model-produced text. We call for the collective efforts from the community to come up with a systematic framework that unifies different metrics, for developing more reliable and natural language generation systems.

## 2 Implementation Details

**Preprocessing.** We utilize three raw datasets: WritingPrompts, WikiText-103, and RealNews. For WritingPrompts, the prompt set has already been well-curated, so we just extracted the first 5,000 prompts (the length may vary) for our generation task. WikiText-103 and RealNews contain many complete texts. For each complete text, we further truncate it corresponding to the first 35 tokens as a prompt. To fairly evaluate the performance of metrics, we also divide text generations according to five predefined length (from 0 up to 1024) intervals for each dataset. Thereby, the human-written texts and model-produced texts used to evaluate the performance of metrics may be generated by different prompts (i.e., unpaired comparison).

**Hyper-parameters.** We have several hyper-parameters during the text generation and evaluation phases. For both conditional and unconditional generation, we preset a random seed integer (32 by default). Furthermore, the maximum length of each text (1024 by default) as well as the batch size (which varies according to GPUs capacity) for perplexity computation have to be determined before automatic evaluation.

## 3 Miscellaneous Details

**Software.** Our experiments were performed on Ubuntu 20.04.1 system with Python 3.9.16. The versions of key Python libraries include: Transformers 4.27.4, PyTorch-CUDA 11.6, PyTorch 1.13.1, Scipy 1.5.4.

**Hardware.** For the text generation task, we use the remote workstation that has two NVIDIA RTX A6000 graphics cards. It should be noted that all models were run in parallel when available.

**Computation time for text generation.** We spent 10 and 25 hours or so obtaining 5,000 text continuations by GPT2-sm, -xl, respectively. OPT-125m, -6.7b cost our GPU resources roughly 11 and 44 hours to output the same number of text continuations, respectively. When it comes to BLOOM-560m, -7b, they took approximately 18 and 48 hours, respectively, to generate 5,000 continuations per task domain.

**Evaluation time for FACE.** Computation time of four FACE metrics for a single pair of references are: $5.96 \times 10^{-8}$ seconds for *SO*, $5.01 \times 10^{-8}$ seconds for *CORR*, $4.53 \times 10^{-8}$ seconds for *SAM*, and $4.29 \times 10^{-8}$ seconds for *SPEAR*, respectively. The cross-entropy, which should be calculated beforehand, takes $5.65 \times 10^{-2}$ seconds. All of the above measurements take place on an AMD Ryzen Threadripper PRO 3995WX 64-Cores CPU (frequency range $\in$ [2200.00MHz, 4308.40MHz]). Users can leverage more advanced GPU resources to perform the whole computation process with a faster speed.

## 4 Additional Experimental Results

### 4.1 Model sizes (generation length)

It should be emphasized that LMs have diverse designs and were pre-trained using different strategies on different datasets, giving them distinct preferences on the generation length. The numbers of text generations in each length interval are summarized in Table 7.

| Domain | Length Interval | GPT2-sm | GPT2-xl | OPT-125m | OPT-6.7b | BLOOM-560m | BLOOM-7b |
|---|---|---|---|---|---|---|---|
| Wiki text | 0-200 | 403 | 485 | 964 | 1522 | 4928 | 803 |
|  | 201-400 | 571 | 672 | 888 | 929 | 61 | 599 |
|  | 401-600 | 251 | 316 | 441 | 417 | 8 | 388 |
|  | 601-800 | 260 | 310 | 268 | 285 | 1 | 316 |
|  | 801-1024 | 3515 | 3217 | 2439 | 1847 | 2 | 2894 |
| News | 0-200 | 750 | 836 | 844 | 1119 | 4978 | 1371 |
|  | 201-400 | 1222 | 1336 | 1220 | 1325 | 20 | 917 |
|  | 401-600 | 824 | 759 | 1194 | 939 | 1 | 628 |
|  | 601-800 | 584 | 678 | 764 | 593 | 0 | 427 |
|  | 801-1024 | 1620 | 1391 | 978 | 1024 | 1 | 1657 |
| Stories | 0-200 | 549 | 745 | 2731 | 3588 | 4924 | 1608 |
|  | 201-400 | 625 | 757 | 715 | 501 | 63 | 688 |
|  | 401-600 | 296 | 404 | 241 | 176 | 9 | 410 |
|  | 601-800 | 241 | 324 | 160 | 95 | 4 | 271 |
|  | 801-1024 | 3289 | 2770 | 1153 | 640 | 0 | 2023 |

Table 7: Domain-specific generation length with respect to different **models** (GPT2/OPT/BLOOM) and **model sizes** (one large model and one small model) using top-$k$ ($k = 50$) sampling corresponding to five continuous length intervals.

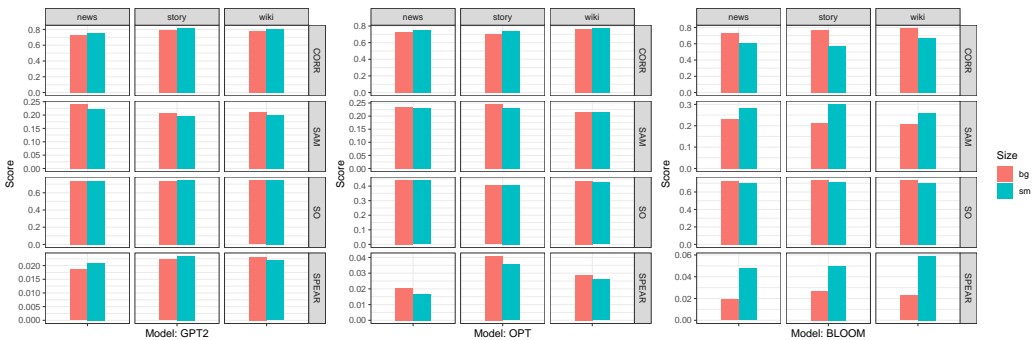

Figure 6: FACE scores of GPT2 (our generated data), OPT, and BLOOM with different model sizes.

To ensure the consistency of our experiments, we run six LMs separately (using their own tokenizers) with the same prompt sets and settings as described in Table 2 to generate 5,000 pieces of continuations in each domain. Besides, we utilize the GPT2Tokenizer to calculate the numbers of continuations for each interval, which allows us to compare FACE scores with other metrics more objectively, as we believe it is unfair to explicitly compare texts of varying lengths. Then, we compute weighted arithmetic mean to evaluate a model in each domain, by $s' = \sum_{i=1}^{n} \frac{m_i}{M} s_i$, where $s'$ denotes the weighted mean; $n$ denotes the number of length intervals; $m_i$ is the number of generated continuations in the length interval $i$; $M = \sum_{i=1}^{n} m_i$, and $s_i$ means a certain metric value in the interval $i$.

Figure 6 conveys a more intuitive representation (via bar plots) of Table 3.

### 4.2 Sampling methods (unconditional generation)

We also carried out experiments on unconditional text generation. Here, the prompt is not required as we generate continuations from a random seed (set to 32 empirically). Four sampling methods, which are greedy decoding, beam search, stochastic beam search, and contrastive decoding, are not involved in this set of experiments.

The results are displayed in Figure 7. The overall trends are same as its conditional counterpart, where the previous quality relationship (maximization-based/temperature-based $\prec$ nucleus $\prec$ contrastive) is satisfied. Yet, it is crucial to note that the advantages of top-$k$ sampling w/o temperature become more obvious compared to the conditional case.

---

[5]`https://github.com/ari-holtzman/degen`

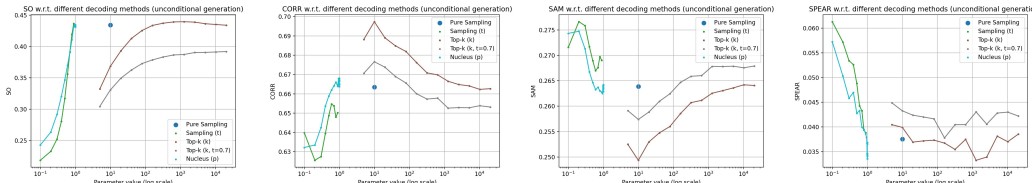

Figure 7: FACE scores (unconditional generation) on original experimental data[5] of *nucleus sampling*. Five sampling (decoding) methods are compared: pure sampling, temperature, top-$k$, top-$k$ with temperature, and nucleus. Note that logarithmic normalization on parameter values as well as an enlarged marker for pure sampling are adopted for better visualization.

| Model | Sampling Method (parameter) | *SO* | *CORR* | *SAM* | *SPEAR* |
|---|---|---|---|---|---|
| GPT2-xl | Nucleus Sampling ($p$=0.95) | 0.481 | 0.821 | 0.191 | 0.359 |
| | Ancestral Sampling | 0.472 | 0.807 | 0.199 | 0.331 |
| GPT2-lg | Nucleus Sampling ($p$=0.95) | 0.480 | 0.819 | 0.193 | 0.356 |
| | Ancestral Sampling | 0.472 | 0.814 | 0.196 | 0.338 |
| GPT2-md | Nucleus Sampling ($p$=0.9) | 0.478 | 0.815 | 0.194 | 0.358 |
| | Ancestral Sampling | 0.462 | 0.813 | 0.197 | 0.310 |
| GPT2-sm | Nucleus Sampling ($p$=0.9) | 0.476 | 0.817 | 0.194 | 0.359 |
| | Ancestral Sampling | 0.468 | 0.816 | 0.195 | 0.319 |

Table 8: FACE results based on MAUVE's original experimental data[6].

## 4.3 Human judgments

Table 8 shows the FACE scores based on the output texts from MAUVE. Each column of FACE scores is used to compute the Spearman's rank correlation coefficient between a specific FACE metric and Bradley-Terry scores (4 model sizes $\times$ 2 sampling methods = 8 scores in total) from one criterion (three criteria correspond to three questions in total).

## 4.4 Choice of estimator model

We examine how different choices of estimator model $m_{\text{est}}$ affect the resulting spectra of cross-entropy. Five input data sources are examined (webtext plus four GPT2 original output datasets), on which four different estimator models are applied: $m_{\text{est}} \in \{$GPT2-sm, GPT2-md, GPT2-lg, GPT2-xl$\}$, resulting in $5 \times 4 = 20$ aggregated spectra curves in Figure 8. It can be found that on the same input data, the spectra from four estimators largely overlap. It indirectly suggests that FACE should be stable across different $m_{\text{est}}$s. We leave the full inspection for future work.

## 4.5 Intuitive interpretation of spectra

As pointed out in Section 4.5, the aggregated spectral shapes from human and different models are nearly identical. A set of higher resolution plots from GPT-xl, OPT, BLOOM and human (webtext) are shown in Figure 9. It can be seen that although the $X(\omega_k)$ has different ranges on $y$-axis, the $x$ coordinates of the peaks and troughs are the same.

## 4.6 Corner Cases

Two examples highlight the difference between our proposed FACE-*SO* and MAUVE in their ability to recognize human-generated and model-generated texts. In the filtered datasets for human judgments, the average values for FACE-*SO* and MAUVE are 0.4738 and 0.9549, respectively. In Case 1, human evaluators noted a high level of similarity between the model-generated text and human text, resulting

---

[6]https://github.com/krishnap25/mauve-experiments

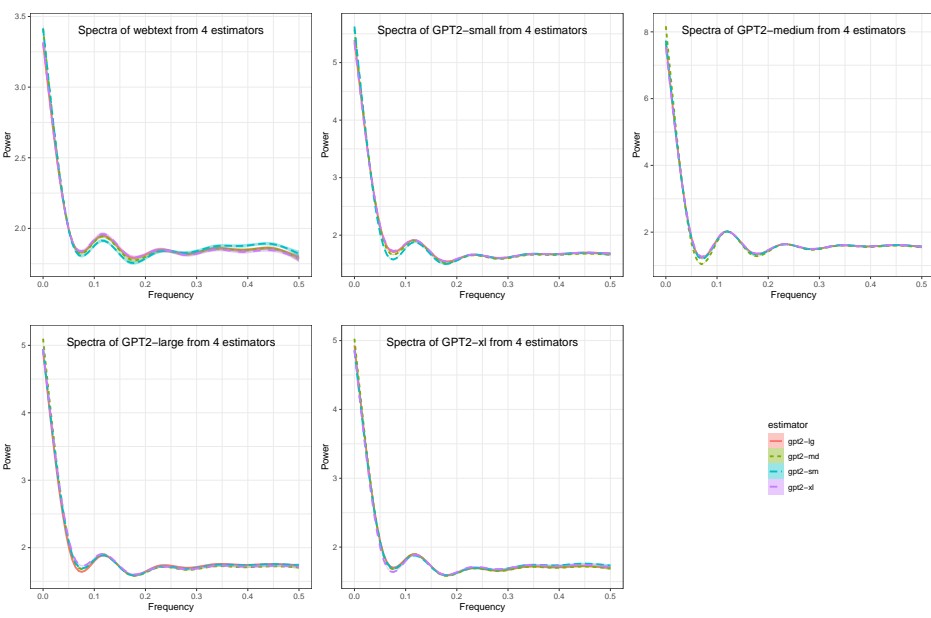

Figure 8: Aggregated spectra (using GAM smoothing) from four estimator models $m_{\text{est}} \in \{\text{GPT2-sm}, \text{GPT2-md}, \text{GPT2-lg}, \text{GPT2-xl}\}$. Inputs are from GPT2 original output and webtext.

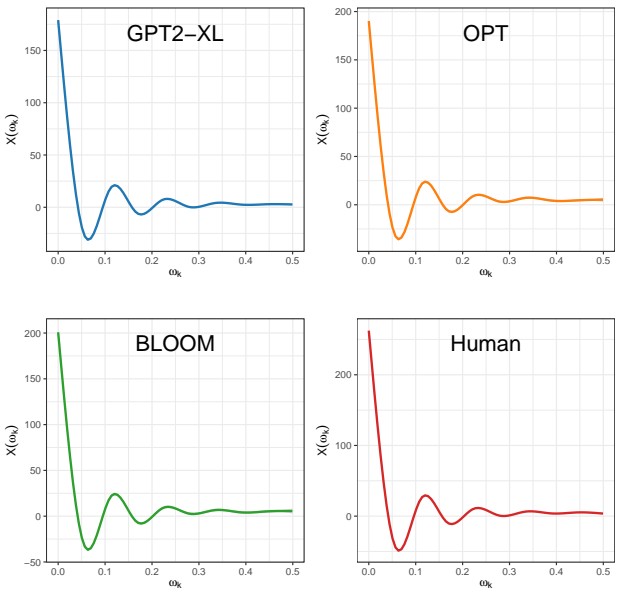

Figure 9: Aggregated spectra for GPT-xl, OPT, BLOOM, and human (webtext).

in ties for human-like, interesting, and sensible aspects. However, the MAUVE score in Case 1 is lower than the average value, while the FACE-*SO* score surpasses its mean. This discrepancy suggests that SO aligns more consistently with human opinions. Conversely, in Case 2, human judgement indicates a significant dissimilarity between the model-generated text and human text, making them easily distinguishable. However, the MAUVE score exceeds its mean, suggesting the two texts are similar to each other. Our FACE-*SO* score is lower than its mean, indicating better alignment with human opinion.

| Case ID | Prompt | Model-generated | Human | MAUVE | SO | Human Judgement |
|---------|--------|-----------------|-------|-------|-----|-----------------|
| 1 | <p>In Arctic Svalbard, there is a vault that might sound like a sci-fi plot device. Completed in 2008 | <p>arding all genetic material from within the world's Arctic Ocean in massive metal containers. The seeds on the vault are extracted in any time zone in the world's North Pole and are safe from theft or re-use at home.</p><p>Here are some of the goals of the Global Seed Vault:</p><ul><li><p>To provide the seeds for the Seed Vault of our inventors: Canaya Berry and Kaleidoscope.</p></li><li><p>To provide the seeds for the Seed Vault of the [Fracked Island from] Israel.</p></li><li><p>To build a mountain pass (with gateway) in southern Greenland</p></li><li><p>To bring purified material to the Arctic.</p></li><li><p>To assemble a home base (with gateway) for our cargo-transport.</p></li></ul><p>In case of rescue need, Alaska government and Navy have the ability to maintain his vessel.</p>, the Global Seed Vault is a remote archive for safegu</p> | <p>arding seeds for thousands of crop varieties. If anything dramatic should happen elsewhere around the world, we want these seeds to be there.</p><p>The vault consists of a giant freezer room bored into a mountain, protected by the bedrock around it and the permafrost above it. But according to a report in The Guardian, the vault experienced an unhappy surprise recently,Äïmelting permafrost in winter.</p><p>The Arctic just experienced its second-warmest winter on record (surpassed only by 2016), and Svalbard saw remarkable temperatures and even rain. In fact, Svalbard averaged more than 4 ⁓∞C above even the 2004-2013 average.</p><p>As a result, meltwater trickled into the seed vault's entrance tunnel before refreezing. The freezer room itself was safe, but the ice in the tunnel had to be chipped out. Hege Njaa Aschim, a spokesperson for the Norwegian government, told The Guardian, "It was supposed to [operate] without the help of humans, but now we are watching the seed vault 24 hours a</p> | 0.9441 | 0.4995 | Ties in Human-like, Interesting, and Sensible |
| 2 | <p>During the Sundance Film Festival, Park City locals are often treated like second-class citizens, unable to get a table at their favorite restaurant ‚Äî or any restaurant ‚Äî because directors</p> | <p>night.</p><p>The Barcelona man, who was his side's hero of the season until Real Madrid got into the game, has not played a competitive game since this season in Spain's La Liga -- he is likely to come off the bench once again for the Copa del Rey fixture on March 6.</p><p>He was in only his sixth game this season as his side beat Estudiantes 3-0 in the last round of knockout games but Pique played big part in the victory for the La Liga side.</p><p>Barcelona Athletic Club's player of the season, Fv"bio CoentrÂ'Eo and Real Madrid's captain Isco feel sorry for his teammate.</p><p>Waving in the colours of his team, Pique, together with the support of the rest of the teammates, scored the goal that led to the Barca's 5-3 aggregate lead in the last four rounds of the Liga.</p><p>"I feel very sorry for him," said Isco, who was out injured to start the match but returned to the squad. "I think that's the</p> | <ul><li></li></ul><p>First-half goals from Marco Asensio and Karim Benzema, coupled with Sunday's 3-1 victory at Camp Nou, saw Madrid canter to a 5-1 aggregate win, securing the club's seventh trophy since Zidane took charge.</p><p>Pique, who was taken off in the second half with an injury and will undergo tests on Thursday, subsequently conceded that Barca are struggling on and off the pitch.</p><p>"In the nine years I've been [at Barca], it's the first time I feel inferior to Madrid," Pique told reporters after the defeat at the Santiago Bernabeu.</p><p>"We're not having our best time as a team or a club. We should all be as close to each other as possible and move forward.</p><p>Gerard Pique could only watch as an impressive Real Madrid cruised to the Super Cup crown.</p><p>"We must live with this defeat and accept that Madrid are better than us [at the moment], but the season's very long and there's room for improvement."</p> | 0.984 | 0.445 | Human texts are strongly more Human-like, Interesting, and Sensible |

Figure 10: Example of two corner cases. For each case, the prompt text, model-generated text, human text, MAUVE and FACE-*SO* scores, as well as the results from human judgments are tabulated.

