# OpenReview forum: "FACE: Evaluating Natural Language Generation with Fourier Analysis of Cross-Entropy"
_NeurIPS.cc/2023/Conference — NeurIPS 2023 poster_

### Official Review · Reviewer_wEMM · 2023-07-06

**Soundness:** 2 fair
**Presentation:** 3 good
**Contribution:** 2 fair
**Rating:** 3
**Confidence:** 4

**Summary:**

In order to distinguish between human-generated text and machine-generated text, the authors propose the use of the periodicity of cross entropy for discrimination. More specifically, they suggest analyzing cross entropy through the Fourier transform.

**Strengths:**

1. This paper is well-written and easy to follow.
2. The experimental section of this paper is fairly comprehensive.  The authors' experimental objects have broadly encompassed the latest open-source large models. Although it lacks large language models like GPT-3.5 (the cross entropy can still be obtained through APIs).



**Weaknesses:**

1. Motivation. The motivation of the paper is not clear, as the authors do not clearly explain why the CE of human language would exhibit periodicity. In the related work section, they briefly mention previous works, but in my view, dialogue tasks are just a specific case of text generation. Overall, skipping the motivation part significantly reduces the soundness of this paper.

2. Method. The authors' method simply involves applying a FFT to the CE sequences, which I believe lacks substantial novelty. Why haven't the authors considered using the information in the frequency domain as input to a deep neural network to incorporate a powerful NN? Why only analyze information in the frequency domain using spectral similarity metrics? Additionally, most of these metrics have already been presented in [1]. Which method would better utilize this information for discrimination? In conclusion, the proposed method by the authors lacks both sufficient contribution and profound insight.

3. Experiments.  In the experimental section, the authors did not compare against sufficient baselines. For instance, could we achieve good results by only training a contrastive model using human-generated text and LLM-generated text? How helpful is the frequency domain information in discriminating texts?

[1] Y. Xu and D. Reitter. Spectral analysis of information density in dialogue predicts collaborative task performance. ACL

**Questions:**

see weakness

**Limitations:**

 the authors adequately addressed the limitations

---

> ### Author Rebuttal · Authors · 2023-08-09
>
> Regarding Weakness 1:
>
> Please read our general response where we addressed the motivation issue.
>
> Regarding Weakness 2:
>
> Actually applying FFT to CE sequences is our biggest innovation. First of all, as we discussed in detail in the general response, applying Fourier analysis on cross-entropy sequence is motivated by two pieces of previous work, Xu et al. (2017) and Dethlefs et al. (2016), in which the periodical patterns of cross-entropy (or surprisal, information density etc.) is discovered (Xu et al., 2017), and speakers are sensitive to the peaks and troughs of cross-entropy in human-machine dialogue (Dethlefs et al., 2016).
>         These evidence all pointed out the potentials of using frequency-domain information to distinguish human and machine languages. To the best of our knowledge, this perspective is never explored before in the community of natural language processing or AI in general.
>
> Regarding the reason of why not using neural networks to take the spectra as input and cast it as a classification problem, we would like to argue that this paper is to answer "whether-or-not", rather than "how" or "how good". Indeed we have positive results on using the spectral features to classify human vs. model-generated languages, and this could potentially lead to a next step of making the most out of the spectral features by adding stronger NN-based models. But so far at this point, we believe that the current content of the paper has done its job in presenting all the promising proof-of-concept results.
>
> Regarding Weakness 3:
>
> Thanks for your review. In our experimental setting, we regard all the other existing open-ended text generation metrics as our baselines, which corresponds to the `metric` column in Table 3, as well as the heading rows in Table 4 and Table 5, respectively. Indeed, leveraging a "trained and tuned" contrastive model can be a reasonable baseline. However, considering all these widely-adopted metrics in open-ended text generation research, it is sufficient to claim that our proposed FACE metrics have explicable consistency compared to previous work and comparable performance compared to the state-of-the-art MAUVE. If needed, we could add the results of our hand-crafted baseline (e.g., evaluation scores from a contrastive model) to existing tables once our paper is accepted.
>
> Speaking of the advantages of using frequency domain knowledge for distinguishing generated texts, one can refer to the vivid analogy which we present to Reviewer mMGf (quoted as follows):
>
> ```
>  the process of transmitting messages between speakers is like delivering cargo between two merchants, who care most about `what` has been delivered, that is, the semantic meanings of words, sentences, ... (whether they make sense or not, etc). However, another important factor that determines the transportation efficiency is `how much`, or, the `weight` of the cargo, which corresponds to the entropy of words. To utter and to understand a high-entropy word/sentence is expensive in terms of cognitive effort needed, and thus the merchants (speaker) need to arrange the cargo's weight in a reasonable way, so that it is neither too tiring for the delivery driver (language production devices) nor too busy for the receiver side (language comprehension devices).
> ```
>
> In short, utilizing the cross-entropy sequences from the frequency spectra of texts can effectively perceive periodic patterns of high/low-entropy words and thus quantify the difference between two families of generated texts without concerning their surface-level or semantic-level representation. Moreover, our approach is computationally efficient as it only requires cross-entropy inferences with a pre-trained language model $m_{est}$, getting rid of the tedious and time-consuming feature-space operations (e.g., encoding texts into feature space before computing the cosine similarity).

---

> > ### Comment · Reviewer_wEMM · 2023-08-14
> >
> > I have read the rebuttal.
> > However, applying FFT to CE sequences is not novely enough. It is so straightforward after reading [1].
> > Additionally, you said "we would like to argue that this paper is to answer "whether-or-not", rather than "how" or "how good".", but the thing is this paper lacks of in-depth analysis of the frequency domain information in discriminating.

---

> > > ### Author Response · Authors · 2023-08-16
> > >
> > > Our work is different from [1] (Xu et al., 2017) in research objectives and also has significant improvement in methods: \
> > >     1) Xu et al. used simple n-gram language models to compute the cross entropy. We are testing neural language models.\
> > >     2) Xu et al. used periodogram method to obtain spectrum, which we proved in our work is not the most appropriate method. Instead, we proposed to use the raw Fourier Transform without applying any commonly used smoothing windows, and will publish the code to better guide future researchers. This part of work is IS NOT trivial although it is hidden from the paper's main content. \
> > >     3) The proposed four metrics of FACE are already an in-depth analysis of the frequency domain information. From our results, we can conclude that most of the spectral difference between human and model is in the low-frequency components. There is space for deeper analysis, which we think should be left to future work.\
> > >     4) The idea of applying FFT to CE sequences only appeared once in the proceedings of ACL, and has never been considered as a way to evaluate natural language generation before (to the best of our knowledge). This, we believe, sufficiently proves the novelty of our work.
> > >
> > >
> > > We think the NeurlIPS community should avoid the tendency of pursuing unnecessary mathematical complexity for the sake of complexity, and be more open to truly innovative works that promote interdisciplinary insights. Mathematician Joseph Fourier's method was invented in 1822, but it was not until 1965 that Cooly and Tucky invented the FFT algorithm. It was not until 1992 that FFT was used in image compression that led to JPEG. It was not until 2012 that FFT was first used in analytical chemistry [2]. Last but not least, it was not until 2023 that FFT was used for distinguishing human and model languages in the era of AI (our work).
> > >     The point we are trying to make here is that the science community should not let the merit of work be blinded by the superficial judgement of novelty solely based on the \emph{age} of method.
> > >
> > > References\
> > > Fernandez-de-Cossio Diaz, Jorge; Fernandez-de-Cossio, Jorge (2012-08-08). "Computation of Isotopic Peak Center-Mass Distribution by Fourier Transform". Analytical Chemistry. 84 (16): 7052–7056

---

> > > > ### Comment · Reviewer_wEMM · 2023-08-16
> > > >
> > > > Xu et al. used simple n-gram language models to compute the cross entropy. We are testing neural language models.
> > > >
> > > > ==> The backbone of language models should not be considered as your contribution. Both your method and Xu's are backbone-agnostic.
> > > >
> > > > Xu et al. used periodogram method to obtain spectrum, which we proved in our work is not the most appropriate method. Instead, we proposed to use the raw Fourier Transform without applying any commonly used smoothing windows and will publish the code to better guide future researchers. This part of work is IS NOT trivial although it is hidden from the paper's main content.
> > > >
> > > > ==> Firstly, using vanilla FFT is not quite hard with some Python packages. You said it is not trivial, but why? You hid it from the main part then how can I judge this part?  Additionally, Xu et al. use periodogram method which is smoothed version of FFT, and your method is vanilla FFT. I don't think it is a NOT TRIVIAL contribution compared with Xu's method.
> > > >
> > > > About the FT part, you don't need to teach me about the history of FT. I am familiar with FFT, and that's why I think your paper is not novel enough.
> > > > In my view, Xu et al. found that we can use the spectrum features to analyze the entropy sequences. And then you apply this idea to the entropy sequences of human-generated text and LM-generated text. Can you call it "truly innovative work"?

---

> > > > > ### Author Response · Authors · 2023-08-16
> > > > >
> > > > > >Firstly, using vanilla FFT is not quite hard with some Python packages. You said it is not trivial, but why? You hid it from the main part then how can I judge this part? Additionally, Xu et al. use periodogram method which is smoothed version of FFT, and your method is vanilla FFT. I don't think it is a NOT TRIVIAL contribution compared with Xu's method.
> > > > >
> > > > > We summarized the reasons of not using periodogram but raw FT instead in the second paragraph after Eq. 2 on page 3 (quoted below). So it is not totally hidden in the paper.
> > > > > ```
> > > > > We compared two methods, periodogram and vanilla FFT. The periodogram approach computes
> > > > > the Fourier transform after applying auto-correlation and time-averaging windows to the signal for
> > > > > de-noising purposes [42]. However, we think de-noising is inappropriate because our “signal” is a
> > > > > time series of cross-entropy, whose value reflects the sampling result at each time step from a large. Auto-correlation or time averaging will remove the distinctiveness of rare tokens. Therefore, we use vanilla FFT and take the real part of X(ωk) to represent the magnitude spectrum for the frequency component ωk, which is written as X(ωk) for brevity.
> > > > > ```
> > > > >
> > > > > It is a non-trivial contribution because we found that cross-entropy (measured in bit) should not be treated as if it is a regular time-domain signals, especially when smoothing techniques are to be applied. Spikes in cross-entropy indicates rare linguistic information, which should not be smoothed out as they are not "noise". So, we actually identified a very special requirement for carrying out Fourier analysis or other spectral analysis on language data, that is, do not take it too easy in removing outliers. This is a distinct improvement over Xu et al. (2017)'s method. Although vanilla FT is "simpler" than periodogram in terms of algorithm complexity, but the decision-making of using the right method that meet specific needs is time-taking and not an easy task. Most importantly, our decision-making process is not a random guess but actually has scientific reasons behind.

---

> > > > > > ### Comment · Reviewer_wEMM · 2023-08-19
> > > > > >
> > > > > > I have read your rebuttal, but I still have concerns regarding the depth of the problem addressed. In our fields, a seemingly simple solution can arise from important insights into a profound or non-trivial problem.
> > > > > > A classic example is the Residual Networks (ResNet), where a technically simple concept solved a fundamental issue of training deep networks to fit identity mappings.
> > > > > >
> > > > > > In contrast, your work appears to address a more direct and straightforward problem, i.e., de-noising is inappropriate, potentially lacking depth or broad relevance. Your efforts appear to focus on feature engineering, leading to the discovery of favorable features. While this is also a valid and potentially valuable approach, it seems to lack the profound difference that characterizes paradigm-shifting works. so I worry that the motivation and solution may limit the innovation and impact of this work.

---

### Official Review · Reviewer_s437 · 2023-07-08

**Soundness:** 3 good
**Presentation:** 4 excellent
**Contribution:** 3 good
**Rating:** 6
**Confidence:** 4

**Summary:**

This paper proposes a new measure of natural language generation (NLG) quality based on similarity between the spectrum of cross-entropy in natural vs. generated text. Fourier Analysis of the Cross-Entropy of language (FACE) is inspired by NLP and psycholinguistic studies suggesting that surprisal is not uniformly distributed in natural text (e.g., content words tend to be more surprising than function words), occurring periodically. For a given generated text, FACE computes a discrete Fourier transform of the token-level cross-entropy sequence (under a separate FACE evaluation LM). Similarity between the vector of frequency magnitudes and that from a randomly selected, natural text corpus are then computed. The paper considers several definitions of FACE metrics, including spectral overlap, cosine similarity, and Pearson/Spearman’s rank correlation coefficients.

LMs from 125 million to over 7 billion parameters are evaluated on NLG of Wikipedia articles, news articles, and stories (with a short prompt of 35 subword tokens provided). Ultimately, FACE is found to be correlated with human judgments of how “human-like”, “sensible”, and “interesting” the generations are. The relationship is not as strong as an existing intrinsic measure, MAUVE. The relative ranking of decoding methods according to FACE agrees with prior works (e.g., greedy decoding < nucleus), as do model size (smaller models produce lower quality generations than larger models).

**Strengths:**

The metric is well-motivated, evaluating whether generated text matches the surprisal statistics of natural text. The algorithm is simple and described sufficiently clearly. FACE is an automatic measure of NLG quality that is, on the face of it, complementary to existing measures. This paper would be of interest to many who work on (large) language models.

**Weaknesses:**

While FACE is motivated by the desire to match surprisal statistics of natural text, it was not clear how different FACE is from existing metrics. Computing correlation between FACE and existing metrics would help alleviate this, as would providing anecdotes of cases with high/low FACE score vs. high/low MAUVE score, for instance.

**Questions:**

Have you also considered the spectrum of hidden LM embeddings rather than cross-entropy, and considered how such a metric might differ from FACE?

**Limitations:**

Yes

---

> ### Author Rebuttal · Authors · 2023-08-09
>
> Regarding Weakness:
>
> Thank you for your review. If needed, we can include examples in our paper once it is accepted. It is important to note that our approach considers the distinctions between human and model-generated languages in terms of cross entropy and periodicity, which sets it apart from most current mainstream metrics. Our research delves into the implicit aspects of human language usage, for example, exploring the periodicity of high/low-frequency words. For more insights, please check our PDF file, which provides corner cases and experimental results demonstrating SO is better on recognising the model-generated and human texts.
>
> Regarding Question:
>
> Our approach is designed based on the theory of FFT (Fast Fourier Transform), making the investigation of periodicity most intuitive and sensible using cross-entropy as inputs. Using hidden embeddings may induce other problems, such as defining the strength of a signal or making analogies. Additionally, our aim is to investigate the periodicity of high/low-frequency words, making the use of cross-entropy more appropriate.

---

> ### Comment · Reviewer_s437 · 2023-08-21
>
> I have read the author rebuttal.

---

### Official Review · Reviewer_mMGf · 2023-07-12

**Soundness:** 3 good
**Presentation:** 3 good
**Contribution:** 3 good
**Rating:** 5
**Confidence:** 4

**Summary:**

This paper proposes a set of metrics based on Fourier Analysis of the estimated Cross-Entropy (FACE) of language. The main idea is to compute the similarity between the spectra of cross-entropy in model-generated texts and human-written texts. Experimental results show that FACE as a computationally efficient metric can scale with model size and reflect the outcomes of different sampling methods for decoding.

**Strengths:**

1. The idea to introduce the spectra of cross-entropy into the evaluation task of open-ended text generation is interesting since it may include some patterns (e.g. periodical patterns) to identify the difference between model-generated texts and human-written texts.

2. This paper is overall well-written and easy to follow.

**Weaknesses:**

1. The proposed method lacks deeper analysis on the spectrum of cross entropy in the evaluation task. The authors only use the spectrum of cross entropy as a feature vector of texts to compute similarities without clearly describing the characteristics of texts it can reflect. This seems like an empirical try without definite intuitions or theoretical supports. In comparison, the features which are commonly used in the existing metrics such as n-gram statistics (in BLEU) and contextual hidden vectors (in BERTScore) intuitively indicate the surface-level and semantic-level representation of texts, respectively.

2. From Table 5, the performance of SO is still worse than that of MAUVE proposed in 2021. I understand that pursuing SOTA is not necessary for each paper. But the authors should provide more insights into the advantages of SO over MAUVE in other aspects.

3. In Section 4.4, the authors mention that they use GPT-2 of different scales to compute the spectra of GPT-2 output data. I wonder whether this setting can introduce potential bias because the cross entropy may be exceptionally low when using GPT-2 to evaluate its own output data from my experience.


**Questions:**

I have included my questions in the weaknesses part.

**Limitations:**

The authors have adequately addressed the limitations.

---

> ### Author Rebuttal · Authors · 2023-08-09
>
> Regarding Weakness 1:
>
> Indeed, in our preliminary research, we conducted a comprehensive analysis of the spectrum of cross entropy and observed its ability to effectively reflect the periodic patterns of high/low entropy words.
> It is important to note that our work is not just an empirical trial; rather, it is theoretically inspired and supported by relevant literature. References 7 and 46 in our research demonstrate that cross-entropy changes periodically in natural language, which presents a potential means to quantify differences in texts.
>
> In daily-life language use cases, we may not explicitly pay attention to word entropy, as our primary task is to convey semantic message.
>         To raise an analogy, the process of transmitting messages between speakers is like delivering cargo between two merchants, who care most about `what` has been delivered, that is, the semantic meanings of words, sentences, ... (whether they make sense or not, etc). However, another important factor that determines the transportation efficiency is `how much`, or, the `weight` of the cargo, which corresponds to the entropy of words. To utter and to understand a high-entropy word/sentence is expensive in terms of cognitive effort needed, and thus the merchants (speaker) need to arrange the cargo's weight in a reasonable way, so that it is neither too tiring for the delivery driver (language production devices) nor too busy for the receiver side (language comprehension devices).
>
> The most relevant previous work in psycholinguistics is from Xu et al. (2017). They pointed out that in natural dialogue, rational speakers should avoid overlapping `peaks` of entropy (cargo weight) to achieve better communication, which can be reflected on the similarity of entropy spectra.
>         It leads to the basic assumption of our study: probably a human speaker (biological being) tends to save effort  by following some periodical patterns in delivering `heavy` words, while a machine speaker does not show such a tendency (because it does not need it).
>
> Regarding Weakness 2:
>
> Thanks! We understand and agree with your perspective on evaluating the quality of generated texts. Indeed, it is a subjective task, and different individuals may have varying opinions, making the state-of-the-art (SOTA) evaluation flexible.
>
> Our approach, which focuses on detecting differences in periodicity and cross-entropy, seems to implicitly and profoundly explore human language usage. By delving into these aspects, our research aims to uncover and understand the nuances of how humans use language, which is an essential and valuable contribution to the field. It provides insights into the intricacies of human language expression and recognition, potentially leading to significant advancements in natural language processing and understanding. If needed, we could add examples once our paper is accepted. For more insights, please check our PDF file, which provides corner cases and experimental results demonstrating SO is better on recognising the model-generated and human texts.
>
> Regarding Weakness 3:
>
> Thank you for pointing this out. In our preliminary research, we tried GPT-2 with three different sizes, and we found that while the absolute values changed, they did not affect the periodicity of the results.
>
> It is true that GPT2-xl will result in cross-entropy of smaller absolute values than GPT2-sm, but this will not affect the periodical patterns of cross-entropy. An analogy is that cross-entropy of the same sequence estimated by different models are the same signal amplified to different amplitudes, but they still have the same period.

---

> > ### Comment · Reviewer_mMGf · 2023-08-21
> > **Response to Rebuttal**
> >
> > Thanks for your rebuttal. Regarding the response to weakness 1, my concern is that there is still a gap between the method based on the spectrum of cross entropy and the NLG evaluation task. Of course, I know that existing works have explored the properties of the spectrum. But they do not focus on NLG evaluation tasks. Since the main contribution of this paper in my view is applying it to NLG evaluation tasks, the authors should surely analyze the characteristics of texts reflected by the spectrum of cross entropy and clearly describe how they benefit the NLG evaluation task, instead of only citing related papers.

---

### Official Review · Reviewer_AtQ2 · 2023-07-24

**Soundness:** 3 good
**Presentation:** 3 good
**Contribution:** 1 poor
**Rating:** 4
**Confidence:** 3

**Summary:**

This paper proposes a new language generation evaluation metric.
Prior work in psycholinguistics has shown that surprisal changes periodically in natural language, with natural utterances displaying moments of high and low surprisal.
This paper thus proposes to evaluate natural language generation models by quantifying how similar its surprisal frequency patterns are to natural language's.
More specifically, this paper proposes to: (1) estimate the surprisal of natural and model-generated text using a separate pretrained language model; (2) get frequency spectra for this surprisals using discrete fourier transforms; (3) compute 4 different metrics of similarity between the frequency spectra of model- and human-generated surprisals.
They then experiment with this metric, showing how it evaluates models of different sizes, and with different decoding strategies.
In a final experiment, they present the correlation between their metric and human judgment scores for 8 gpt2-based language generation systems (small, medium, large, xl with either ancestral or nucleus sampling); this experiment shows that while the proposed method does better then some prior metrics (e.g. self-bleu) it produces worse correlations than mauve.


**Strengths:**

I found this paper quite interesting.
The motivation was relatively clear and the proposed metric is well-motivated.
In particular, operationalising a psycholinguistic hypothesis of what consists human-like text, and then using it to evaluate language generation systems seems like a promising approach.
Further, I appreciate the idea of using Fourier transforms to analyse the frequency spectra of information/surprisal in text.

**Weaknesses:**

I believe that the evaluation part of the paper could be improved:
* First, section 4.1 discusses how the proposed metric evaluates models of different sizes. (They generate text while prompting models with a few initial tokens from sentences of three datasets.) These results, however, seemed confusing to me; sometimes smaller or larger models are better with no clear explanation, and the main comparison point used is whether or not the proposed metric agrees with mauve. If mauve was to be considered a gold standard, however, we would not need a new metric.
* Second, section 4.2 evaluates the impact of decoding strategy on the evaluated scores. In these experiments, the authors (coherently) find that their proposed metric always evaluates contrastive decoding as the best strategy. How their evaluation metric fares when comparing other decoding strategies (e.g. nucleus vs ancestral sampling), however, is less clear. Further the table does not show any scores for human text, which I believe could work as an interesting sanity test. These could be computed by evaluating the proposed metric using half of this dataset against its other half.
* Third, the correlations with human judgement scores in section 4.3 are worse than mauve's. While I do not believe a paper needs to have state-of-the-art scores to be published, the paper does not put forward other reasons why it should be accepted besides these correlations, and treats these negative results as positive in its discussion.

In summary, this paper focuses on how its proposed evaluation metric produces good scores of what is human-like text, but does not demonstrate to be better than mauve at this. Further, it does not offer any other justifications (besides being a good metric of human-like text) for why one should use it. Together, this makes me think the impact of this paper might be quite limited.

Adding a longer and more detailed comparison between this proposed metric and previously proposed ones could help improve this paper's impact.


**Questions:**

Questions:
* Which model was used to compute $m_{\text{est}}$ in the experiments?
* Line 21 cites Piantadosi (2014) for the claim that "For example, Zipf’s law can be used to distinguish between human and model distributions." I don't think this is a conclusion of Piantadosi (2014), however. Could you clarify where in the paper he reaches that conclusion? If this is about their comparison with "random typing models", those are qualitatively different from language models. When examining proper language models, Meister et al. (2021) reach the opposite conclusion (that language models follow a similar rank-frequency relationship to natural language).

Larger Suggestions:
* From the paper's text in section 2.2, I interpret that the discrete Fourier transform operates in a single sentence at a time, and thus the proposed metric was developed to compare the spectra of two sentences; not of two corpora. Figure 1, however, implies the Fourier transform takes as input all sentences in a dataset at once. Explaining section 2.2 in more detail could be helpful.
* Although the authors do discuss this in their paper, I believe the word cross-entropy is not accurate to describe what is being measured here. The word surprisal is the correct one. The authors themselves note this in line 62, but decide to use the term "cross-entropy" anyway because it was used this way before, as in, e.g. Genzel and Charniak (2002). Personally, I do not believe this to be a good reason—it's not because prior work used the wrong terminology that you should propagate it.

Smaller Suggestions:
* Line 156 states that Mauve "straightforwardly computes" the similarity of the model- and human-text distributions. However, computing this divergence is actually intractable (starting from the fact that human-text distributions are unknown), and so this is not actually straightforward. They just approximate this using clusters of word embeddings. I’d rewrite this as “attempt to compute” or “estimate”.
* Line 158 states that reference-based metrics are suited for close-ended generation settings, putting Mauve in that group. Mauve, however, was actually developed to analyse open-ended generation settings.
* Figure 4 is too small. At the current scale this figure is unreadable.

Meister et al. (2021). Language Model Evaluation Beyond Perplexity.


**Limitations:**

I believe the paper doesn't mention at any point which language their data is in (i.e., English) and that most of the cited psycholinguistics research is English-centric (or at least Indo-European-centric). Addressing that as a potential source of limitation is important.

---

> ### Author Rebuttal · Authors · 2023-08-09
>
> Regarding Weakness 1:
>
> Regarding the results from various model sizes, there are indeed some inconsistent cases where small models out-perform big ones. The data from our current experiments are insufficient to explain this inconsistency, but we are planning a more complete future study to address this. The reason for comparing with MAUVE, however, is not that we treat it as a gold standard, but rather as a means to prove the validity of FACE -- in the worst case, the new metric will be totally uncorrelated with existing ones (such as MAUVE), which did not happen in our study.
>
> Regarding Weakness 2:
>
> The main purpose of having Table 4 in the paper is to show the advantage of FACE in reliably identifying the superior decoding strategies, which, as shown in the previous work (Li et al., 2022), should be contrastive decoding. We admit it is possible that conrtastive decoding may not be the optimal strategy under some conditions, and so we agree with the reviewer that a thorough comparison on other strategies is a plus.
>         Just in case we understand it correctly, if the reviewer suggests that we should look into how FACE scores change when we change the $k$ in top-k sampling from 10 to 640 continuously, or to change the $p$ from 0.1 to 0.995 (as was done in Holtzman et al. 2020), then we agree that these more thorough experiments can better reveal the performance of our metrics.
>         Holtzman et al (2020), i.e., the nucleus sampling paper provided a complete output dataset with a various range of sampling parameters, so we would love to include them in our future investigation.
>
> We appreciate the suggestion of using half the human data to conduct a sanity test. We have conducted the sanity test and found that human text data have better FACE-SO, -CORR, and -SAM scores, which prove the validity of FACE. Please find the results in the attached PDF document in our general response.
>
> Regarding Weakness 3:
>
> We recomputed the correlations with human judgement scores to keep those pairs in which there are exactly one item from human and the other item from model (i.e., a subset of data used for the analysis in Table 5 in section 4.3), and found that FACE-SO has stronger correlation than MAUVE in 2 out of the three dimensions (see table below; full results are in the attached PDF in the general response)
>
>                 | MAUVE | SO
>     Human-Like | 0.214 | 0.357
>     Interesting | 0.667 | 0.524
>     Sensible    | 0.706 | 0.995
>
> It indicates that our metric is better than MAUVE in distinguishing human from model languages, rather than between different models. It also leads to the conclusion that Fourier analysis of cross-entropy can identify some hidden difference between human and model generated text, which is highly likely due to the cognitive capability of human beings in language production.
>
> Regarding Question 1: Which model was used to compute $m_{est}$ in the experiments?
>
> GPT2-small (345M parameters) is used.
>
> Regarding Question 2 about Piantadosi (2014)'s work
>
> Thanks for pointing out this inaccurate citation. It should be corrected to Holtzman et al. (2020), who used the Zipfian coefficient $s$ to compare the distribution in a given text to a theoretically perfect exponential curve, where $s = 1$. Here Holtzman et al. cited Piantadosi (2014). After checking the original Piantadosi paper, we believe the $s$ coefficient referred by Holtzman should be the $\alpha$ in Piantadosi (2014). So, technically, the method of using Zip'f coeffient to distinguish human and machine was first introduced by Holtzman (2020). Please correct us if there is our understanding here is wrong.
>
> It is interesting to read Meister et al. (2021)'s results, which can be potentially useful to our future work.
>
> Reference: A. Holtzman, J. Buys, M. Forbes, and Y. Choi. The Curious Case of Neural Text Degeneration. In Proc. of ICLR, 2020
>
> Regarding Larger Suggestion 1:
>
> It is true that FFT operates on a single sentence at time, and then FACE metrics are computed between two sentences. What Figure 1 shows is the process: we apply FFT to each and every sentence in the corpus, and FACE metrics are computed pair-wise between two shuffled corpora.
>
> Regarding Larger Suggestion 2:
>
> Thanks for pointing this out. Yes, `surprisal` is accepted more broadly in psycholinguistics, but we believe `cross-entropy` (CE) is also compatible here, because `cross` indicates CE measures the difference across two distributions, ground-truth and predictions. The way estimate surprisal also needs the use of a well trained model, and to run predictions with the model. The prediction at each word position is a set of probabilities whose size is the entire vocabulary, among which only one probability is picked based on the ground-truth. Thus, we believe the surprisal itself in nature is the cross-entropy between input sentence and the model. Anyway, it is useful and interesting discussion here.

---

> > ### Comment · Reviewer_AtQ2 · 2023-08-15
> > **Response to Authors**
> >
> > > Regarding Weakness 1:
> > >
> > > The reason for comparing with MAUVE, however, is not that we treat it as a gold standard, but rather as a means to prove the validity of FACE -- in the worst case, the new metric will be totally uncorrelated with existing ones (such as MAUVE), which did not happen in our study.
> >
> > Are you saying that the comparisons between Mauve and FACE are intended only to sanity check the new metric? Section 4.1 is the longest results section in the paper. In my opinion, it doesn't read as a sanity check at the moment.
> >
> > > Regarding Weakness 2:
> > >
> > > We appreciate the suggestion of using half the human data to conduct a sanity test.
> >
> > Thanks for adding these new experiments! I think they are quite interesting.
> > As an additional suggestion (either for CR, in case the paper is accepted, or for a re-submission, in case it is not), Pimentel et al. (2023) recently analysed Mauve and in their Section 6.2 there are a number of experiments (also using only human text data) which would be interesting here to show the potential of the proposed FACE metric. They show, for instance, that removing articles from a text does not significantly impact Mauve, but I guess it should affect FACE, right?
> >
> > > Regarding Weakness 3:
> >
> > It is not clear to me exactly what is being measured in this experiment. What is the correlation being taken across? Is it still across the GPT-2 models? Or are you evaluating sentence-level quality directly now?
> >
> > > Regarding Question 1: Which model was used to compute $m_est$ in the experiments?
> > >
> > > GPT2-small (345M parameters) is used.
> >
> > Do you know how results change with the chosen model? Recent work in psycho-linguistic has shown that GPT-2 small has more predictive power over human reading times than larger and better models (e.g., GPT-2 XL or GPT-3; see Shain et al 2022. or Oh et al. 2023). But I would be curious to know whether that would also be the case for language generation evaluation.
> >
> > # References:
> >
> > Pimentel et al. (2023). On the Usefulness of Embeddings, Clusters and Strings for Text Generator Evaluation.
> > Shain et al. (2022). Large-scale evidence for logarithmic effects of word predictability on reading time.
> > Oh et al. (2023). Why does surprisal from larger transformer-based language models provide a poorer fit to human reading times?

---

> > > ### Author Response · Authors · 2023-08-16
> > >
> > > >Are you saying that the comparisons between MAUVE and FACE are intended only to sanity check the new metric? Section 4.1 is the longest results section in the paper. In my opinion, it doesn't read as a sanity check at the moment.
> > >
> > > Thanks for your follow-up enquiry. In Section 4.1, our aim is not to solely perform the sanity check on FACE metrics. Instead, we focus on validating the effectiveness of our proposed metrics, concerning whether they scale well with model size. To this end, we design our own experimental protocol to evaluate text generations obtained from paired large-small models across three LLM families in a domain-specific manner, which is more comprehensive than MAUVE's.
> > >     We did not regard MAUVE as the gold standard in Section 4.1 because it is only one of the seven metrics (Diversity, Coherence, Zipf, ...) being investigated. Here, MAUVE acts more like a ``good-enough'' reference for us to observe the correlations.
> > >
> > > We admit some of the results did seem confusing. We conjecture there are two reasons to the contradictory results in GPT-2 columns: First, unlike the text generations obtained from OPT and BLOOM models where we downloaded pre-trained LLM and ran inferences with our self-tuned hyperparameters, the generated texts from GPT-2 models are retrieved directly from OpenAI official repository. Second, as you have pointed out, GPT-2 small has been proved to possess more predictive power than GPT-2 XL or GPT-3 counterparts, which probably explains some of the opposite results in the GPT-2 columns (i.e., S better than L).
> > >
> > > We hope the above comments addressed your confusion. We will add deeper and more extensive explanation to make clearer statements once our manuscript is accepted.
> > >
> > >
> > > >As an additional suggestion (either for CR, in case the paper is accepted, or for a re-submission, in case it is not), Pimentel et al. (2023) recently analysed Mauve ... for instance, that removing articles from a text does not significantly impact Mauve, but I guess it should affect FACE, right?
> > >
> > > Thanks for pointing Pimentel et al. (2023) to us. The stability of metrics is an interesting direction to test in our future work. We guess there must be some threshold of text amount that must be met in order to obtain a reliable FACE score. From our current experiments, more data will naturally result in smaller standard deviations in scores, but also this trend scales is yet to be studied.
> > >
> > > >It is not clear to me exactly what is being measured in this experiment. What is the correlation being taken across? Is it still across the GPT-2 models? Or are you evaluating sentence-level quality directly now?
> > >
> > > As a brief summary of the experiment, we are measuring the alignment between human judges and computational metrics on assessing model's generation capability. The evaluation data contain two input text columns $a$ and $b$, and at least one of them is generated from a model, or both of them are from models (of different sizes). Then in the third column $h$, human judges from crowd-source platform give rates on which one among $a$ and $b$ is better. The last column $m$ records which one among $a$ and $b$ gets higher score according to the metric (MAUVE, FACE, or others).
> > > Therefore, higher alignment between columns $h$ and $m$ indicates that the metric aligns better towards human judges. The Bradley-Terry (BT) scores reported in Table 4 measures this alignment.
> > >
> > > In our original results, MAUVE has overall higher BT scores than FACE. We think this is because most of the $<a,b>$ pairs are both from models, and FACE is not as good in distinguishing model from model as MAUVE does.
> > >     In our follow-up experiments, however, we found that FACE has higher BT scores than MAUVE after keeping only those $<a,b>$ pairs in which one source is human and the other is model. We think this is an exciting evidence indicating better alignment between FACE and human judges. It further potentially indicates the profound difference between human- and model- generated text in spectral space.
> > >
> > > Regarding your question \emph{is it still across the GPT-2 models}: Yes. The evaluation dataset is provided by MAUVE's authors, which includes textual data from human-generated and four GPT-2 sizes times two sampling methods (8 combinations in total).
> > >
> > > Regarding your question whether it is sentence-level evaluation or not: No, the evaluation isn't focused solely on sentence-level quality. Within the dataset, the text columns $a$ and $b$ consist of multiple sentences, making it a more comprehensive representation of language usage and quality.
> > >
> > >
> > > >Do you know how results change with the chosen model?
> > >
> > > The output of FACE metrics do not change with the chosen estimator model $m_{est}$. We conjecture this is because $m_{est}$ only affect the magnitudes of cross-entropy (e.g., GPT2sm has larger cross-entropy than GPT2xl), but does not the innate periodical patterns of cross-entropy.

---

### Official Review · Reviewer_v6cq · 2023-07-25

**Soundness:** 3 good
**Presentation:** 3 good
**Contribution:** 3 good
**Rating:** 6
**Confidence:** 4

**Summary:**

This paper proposes a set of metrics to measure the distance between model-generated and human-written languages. Specifically, this paper uses FFT to analyze the cross-entropy sequences of the language data.

**Strengths:**

1. This new metric is efficient. Given the fact that our models are getting exponentially bigger, it is essential that we do not waste energy during evaluation.
2. This new metric correlates well with human judgment, and is statistically sound.

I personally really like the authors' attempt to interpret the metric. Understanding the why is sometimes much more important than understanding the how.

**Weaknesses:**

1. The related work on psycholinguistic motivation is limited. Entropy is also a popular metric in computational linguistics, which is probably worth citing.
2. The model size categorization seems to be very coarse.

**Questions:**

1. Could the authors be more specific about their motivations for using spectral similarity as a metric?

**Limitations:**

This paper is a good step towards addressing some of the problems brought by generative AI.

---

> ### Author Rebuttal · Authors · 2023-08-09
>
> Regarding Weakness 1: The related work on psycholinguistic motivation is limited. Entropy is also a popular metric in computational linguistics, which is probably worth citing.
>
> Thanks for pointing this out. We will read through our paper in detail to include more comprehensive citations once our work gets accepted.
>
> Regarding Weakness 2: The model size categorization seems to be very coarse.
>
> Indeed, in our experimental setting, we only consider "polarized" groups where there is an extremely small language model having fewer parameters paired with a fairly large language model owning more than ten times the parameters for each model family, with the aim of showing FACE's scalability with model sizes. Adopting a finegrained model size categorization (e.g., five language models with linearly increased model sizes in each family) is definitely helpful to demonstrate the robustness of our proposed metrics. Nevertheless, we believe our current setting is sufficient for validating how FACE scales with model sizes since we have already taken two models with a "stark" contrast in the number of parameters across three LLM families into account. We have stated that larger models (with more than 100 billion parameters) need to be included for future work in line 299, but a more finegrained categorization strategy is also worth considering.
>
> Regarding the question "Could the authors be more specific about their motivations for using spectral similarity as a metric?"
>
> Please read our general response about the research motivation.

---

> > ### Comment · Reviewer_v6cq · 2023-08-10
> >
> > I have read the rebuttal.

---

### Author Rebuttal · Authors · 2023-08-09

We would like to thank all the reviewers for providing the useful feedback for further improving our paper. We notice that some reviewers suggest us to strengthen the motivation part, especially the reasons of using Fourier analysis on the cross-entropy sequence of language. We believe that the motivation is sufficiently done in the Introduction and Related Work sections, but probably due to that some of the reviewers are not from psycholinguistics background and thus lack the bigger context, so here we would like to compose a general response to better address this paper's motivation with as plain language as possible.

The most relevant two pieces of work from previous computational psycholinguistics studies are Xu et al. (2017) and Dethlefs et al. (2016). Xu's main finding is that the cross entropy of dialogue utterances has periodogical patterns, which can be used to predict the task success. Dethlefs' results tell about that human speakers' experiences are sensitive to the peaks and troughs in utterances from a human-machine dialogue system. Combining their findings, we distilled our main assumption/idea: capturing the periodical patterns of cross-entropy with FFT, and then test whether the spectral features is an indicator of "good" and "natural" language. It is following this assumption that we complete the entire study step by step.

As for why cross-entropy works, if we are to give a brief explanations of our assumption/, we would say it reflects human beings cognitive capability of producing and comprehending languages. This capability naturally fluctuates in time as humans adapt to information dynamically based on its cognitive load, which produces the temporal periodical pattern of cross-entropy in language. Machines/models, on the other side, do not present such a pattern because it does not have the cognitive load issue. We believe the current way models are trained i.e., via maximum-likelihood estimation-based learning of which words produced at when, does not capture the dynamics of cognitive load behind the curtain.
    Perhaps reviewers outside the field will find our response to Reviewer mMGf an interesting explanation on why it works to use cross-entropy for evaluating language quality.

To sum, this paper is to answer "whether-or-not", rather than "how" or "how good". Fortunately we have positive results on using the spectral features to tell apart human vs. model-generated languages, and this could potentially lead to a next step of making the most out of the spectral features by adding stronger predicting models (as one of the reviewers suggest). But so far at this point, we believe that the current content of the paper has done its job in presenting all the promising proof-of-concept results.

In the attached PDF document, we included three experiments as suggested by the reviewers: 1) Sanity test of FACE by splitting human text data in half. 2) Corner cases of text whose MAUVE scores are higher than FACE, but actually did not read well; some qualitative analysis is included. 3) FACE-SO show higher correlation with human judgement scores than MAUVE, when we use a more reasonable subset of evaluation data.

---

### Decision · Program_Chairs · 2023-09-21

**Decision:**

Accept (poster)

**Comment:**

The paper proposes an evaluation metric for natural language generation. The proposed methodology is novel. The evaluation section is rather weak and several reviewers have highlighted it. The paper can easily be improved and strengthened based on the detailed comments and discussion done with Reviewer AtQ2 and mMGf. I intend to accept the paper since the use of Fourier analysis is valuable for the community. I would encourage authors to incorporate the comments of the reviewers in the final version of the paper.